# Mitochondrial DNA Dysfunction in Cardiovascular Diseases: A Novel Therapeutic Target

**DOI:** 10.3390/antiox14091138

**Published:** 2025-09-21

**Authors:** Mi Xiang, Mengling Yang, Lijuan Zhang, Xiaohu Ouyang, Alexey Sarapultsev, Shanshan Luo, Desheng Hu

**Affiliations:** 1Department of Integrated Traditional Chinese and Western Medicine, Union Hospital, Tongji Medical College, Huazhong University of Science and Technology, Wuhan 430022, China; d202381945@hust.edu.cn (M.X.); d202281868@hust.edu.cn (M.Y.); 2012xh0937@hust.edu.cn (L.Z.); d202482139@hust.edu.cn (X.O.); 2Russian-Chinese Education and Research Center of System Pathology, South Ural State University, 454080 Chelyabinsk, Russia; a.sarapultsev@gmail.com; 3Institute of Immunology and Physiology, Ural Branch of the Russian Academy of Science, 620049 Ekaterinburg, Russia; 4Institute of Hematology, Union Hospital, Tongji Medical College, Huazhong University of Science and Technology, Wuhan 430022, China; 2018XH0258@hust.edu.cn; 5China-Russia Medical Research Center for Stress Immunology, Union Hospital, Tongji Medical College, Huazhong University of Science and Technology, Wuhan 430022, China

**Keywords:** mitochondrial DNA, cardiovascular disease, cellular biology, therapeutic strategy

## Abstract

Cardiovascular diseases hinge on a vicious, self-amplifying cycle in which mitochondrial deoxyribonucleic acid (mtDNA) dysfunction undermines cardiac bioenergetics and unleashes sterile inflammation. The heart’s reliance on oxidative phosphorylation (OXPHOS) makes it exquisitely sensitive to mtDNA insults—mutations, oxidative lesions, copy-number shifts, or aberrant methylation—that impair ATP production, elevate reactive oxygen species (ROS), and further damage the mitochondrial genome. Damaged mtDNA fragments then escape into the cytosol, where they aberrantly engage cGAS–STING, TLR9, and NLRP3 pathways, driving cytokine storms, pyroptosis, and tissue injury. We propose that this cycle represents an almost unifying pathogenic mechanism in a spectrum of mtDNA-driven cardiovascular disorders. In this review, we aim to synthesize the pathophysiological roles of mtDNA in this cycle and its implications for cardiovascular diseases. Furthermore, we seek to evaluate preclinical and clinical strategies aimed at interrupting this cycle—bolstering mtDNA repair and copy-number maintenance, reversing pathogenic methylation, and blocking mtDNA-triggered innate immune activation—and discuss critical gaps that must be bridged to translate these approaches into precision mitochondrial genome medicine for cardiovascular disease.

## 1. Introduction

Mitochondria are highly specialized, double-membrane organelles with cell- and tissue-specific morphology, dynamics, and functions. They play a central role in metabolism and signaling pathways, including energy generation, calcium homeostasis, ROS production, heme synthesis, steroidogenesis, programmed cell death, and metabolite catalysis [1,2,3]. Each mitochondrion contains multiple copies of circular mitochondrial DNA (mtDNA), organized within nucleoids in the inner mitochondrial matrix, and characterized by its own transcriptional machinery [2,4,5]. Animal tissue cells contain thousands of mtDNA copies spread out over hundreds of mitochondria, encoding 13 polypeptides for the respiratory chain complexes, 22 transfer ribonucleic acids (tRNAs), and 2 ribosomal RNAs (rRNAs), as well as enzymes needed for their transcription and translation [2,3,5,6,7]. Notably, mtDNA copies often exceed what is strictly necessary for oxidative phosphorylation, suggesting additional roles in mitochondrial signaling or immune regulation [2]. Like nuclear DNA (nDNA), mtDNA is vulnerable to mutations and exogenous or endogenous damage, which can drive mitochondrial dysfunction and human pathologies [4,8,9,10,11].

Given the heart’s absolute dependence on OXPHOS, mtDNA damage naturally manifests in cardiac syndromes—either as primary mitochondrial disease or as part of multisystem disorders. Clinically, the vicious cycle of bioenergetic failure and sterile inflammation manifests across cardiovascular diseases. In hypertension [12,13,14,15] and atherosclerosis [16,17], mtDNA mutations have been widely reported, and oxidized mtDNA accelerates vascular senescence and exacerbates plaque inflammation. In myocardial ischemia, damaged mtDNA is released and triggers severe inflammatory responses, exacerbating tissue damage [18,19,20,21,22,23,24,25,26]. In heart failure, mtDNA lesions and depletion are involved in critical ATP deficits and worsen contractile dysfunction [27,28]. In diabetic cardiomyopathy, fatty acid-driven mtDNA release provokes cardiomyocyte pyroptosis via the cGAS–STING pathway [29,30]. Arrhythmias, too, are associated with oxidative lesions and mutations in mtDNA [31,32]. Thus, mtDNA has emerged as a critical target for mechanistic understanding and therapeutic innovation in cardiovascular medicine. Elucidating the molecular pathways linking mtDNA integrity to disease progression may open new avenues for biomarker development and precise interventions.

In this review, we synthesize the physiological and pathological features of mtDNA, propose a unified “vicious cycle” theory linking mtDNA dysfunction to cardiac bioenergetic collapse and sterile inflammation, and highlight emerging therapeutic strategies and remaining challenges. Our goal is to inform the development of systematic, mtDNA-targeted interventions that mitigate myocardial injury, prevent heart failure, and reduce mortality.

## 2. Overview of mtDNA

The origin of mitochondria as an ancient archaeal endosymbiont endows the organelle with its own double-stranded, circular genome [33]. This autonomous genome retains the machinery needed to replicate, transcribe, and translate its genes, thereby allowing mitochondria to self-regulate critical functions such as energy production, signaling, and apoptosis (Figure 1).

### 2.1. Structure of mtDNA

MtDNA exists as a closed, circular duplex with a purine-rich heavy (H) strand and a pyrimidine-rich light (L) strand [3,4]. It contains 37 genes-28 on the H-strand and 9 on the L-strand-that are directly or indirectly involved in ATP synthesis via oxidative phosphorylation [3,5,34]. Unlike the nDNA counterpart, mtDNA is extraordinarily gene-dense (~93% coding), with overlapping reading frames and only one major non-coding region, the displacement loop (D-loop), which harbors the origin of replication and promoters for transcription [3].

Rather than being wrapped around histones, mtDNA is packaged into 100-nm nucleoids through binding to the nuclear-encoded mitochondrial transcription factor A (TFAM) [4,35]. TFAM is imported into mitochondria via an N-terminal mitochondrial targeting peptide that is cleaved upon entry to form mature TFAM. In the mitochondrial matrix, TFAM coats mtDNA to stabilize its structure and binds the D-loop to coordinate the bidirectional transcription from the light-strand promoter (LSP) and two heavy-strand promoters (HSP1, HSP2) [4,36,37].

### 2.2. MtDNA Replication, Transcription, and Translation

MtDNA replication occurs continuously, independent of the cell-division cycle, and is entirely governed by nuclear-encoded factors [3]. In eukaryotes, mtDNA replication takes place within a “replisome” complex whose catalytic core is DNA polymerase γ (POLγ), accompanied by two accessory subunits [3,5]. This machinery also includes the mitochondrial single-stranded DNA-binding protein (mtSSB), which stabilizes unwound DNA and enhances POLγ activity; the Twinkle helicase/primase; various topoisomerases; and TFAM, which may protect mtDNA from oxidative damage [3]. Mammalian mtDNA replicates mainly by a strand-displacement mechanism: synthesis of the heavy (H) strand initiates at the D-loop origin and proceeds unidirectionally until exposing the light (L)-strand origin, at which point L-strand synthesis begins in the opposite direction. Both strands elongate asynchronously until two daughter molecules are complete [38,39].

Transcription of mtDNA is driven by three promoters: the light-strand promoter (LSP) and two heavy-strand promoters (HSP1, HSP2) [37]. Nuclear-encoded mitochondrial RNA polymerase (POLRMT) forms a pre-initiation complex with TFAM and TFB2M at the D-loop, melting DNA to expose each promoter [40,41]. From HSP2 and LSP, genome-length polycistronic transcripts are produced; the H-strand is additionally transcribed from promoter HSP1 to generate a shorter transcript consisting of two rRNAs and two tRNAs [42]. Transcript elongation is carried out by POLRMT, with assistance from the mitochondrial transcription elongation factor, while termination is regulated by mitochondrial transcription termination factor 1 (MTERF1) [3].

The mitochondrial ribosomes utilize a unique set of translation initiation, elongation, and termination factors, culminating in a translation cycle completely different from the bacterial and cytosolic ones [43,44]. Initiation factor mtIF-3 dissociates ribosomal subunits to allow assembly of the initiation complex, aligning the start codon of mRNA on the small subunit’s P-site [3,45]. Peptide elongation is mediated by nuclear-encoded mitochondrial elongation factor Tu (tRNA delivery) and mitochondrial elongation factor G (mRNA–tRNA translocation and ribosome recycling) [3,46,47,48,49,50]. Termination relies on mitochondrial release factor 1 (mtRF1), which recognizes stop codons and catalyzes peptide release [3,45].

### 2.3. MtDNA in OXPHOS System

The mitochondrial OXPHOS system serves the fundamental purpose of ATP transduction by oxidizing fuel molecules via the respiratory electron transport chain (ETC) [7]. The mammalian ETC comprises five multimeric complexes (Complexes I–V) embedded in the inner mitochondrial membrane [51]. These complexes harness energy released from electron transfer from NADH or FADH_2_, largely generated by the tricarboxylic acid cycle-to pump protons across the inner mitochondrial membrane, creating a proton-motive force that Complex V uses to synthesize ATP [51,52,53].

Although the majority of OXPHOS subunits are encoded in the nucleus, translated in the cytosol, and imported into mitochondria [7,37,51], mtDNA encodes 13 indispensable polypeptides that belong to four of the five complexes (I, III, IV, and V) [3,7,37,54]. In total, 92 structural OXPHOS subunits-13 mtDNA-encoded and 79 nDNA-encoded-assemble into a highly coordinated machinery [3,34].

Complex I contains 44 subunits (7 mtDNA-encoded ND1-ND6, ND4L, and 37 nDNA-encoded), Complex II is entirely nDNA-encoded (4 subunits), Complex III includes one mtDNA-encoded cytochrome b plus 10 nDNA-encoded subunits, Complex IV comprises three mtDNA-encoded (COI–COIII) and 11 nDNA-encoded proteins, and Complex V (ATP synthase) contains two mtDNA-encoded (ATPase6, ATPase8) and 17 nDNA-encoded subunits [3,34,55,56]. Together, nDNA- and mtDNA-encoded components orchestrate efficient electron flow and prevent accumulation of harmful assembly intermediates [51].

## 3. MtDNA Dysfunction in Cardiovascular Diseases

In the heart, mitochondrial oxidative metabolism supplies approximately 90% of cellular ATP, underlining its centrality to cardiovascular function [57]. Cardiac mtDNA governs both the mitochondrial lifecycle and the integrity of the oxidative phosphorylation system [58]. Excess ROS in cardiomyocytes and endothelial cells inflicts irreversible mtDNA damage, leading to mutations that further impair OXPHOS and disrupt mitophagy. Such dysfunction precipitates the escape of mtDNA and mitochondrial proteins into the cytosol, activating innate immune pathways and driving tissue injury [57,58]. These intra- and extra-mitochondrial mtDNA–mediated processes (Figure 2 and Figure 3) highlight mtDNA as a compelling therapeutic target. Table 1 provides representative examples of distinct cardiovascular pathologies driven by mtDNA abnormalities, summarizing their underlying mechanisms and the supporting evidence.

### 3.1. MtDNA Oxidative Damage

Any damage, especially oxidative damage to mtDNA can alter mitochondrial transcripts and impair oxidative phosphorylation [59]. Mitochondria are significant sources of ROS in cells [60,61] and mtDNA is a major target for ROS-mediated damage due to its proximity to the principal ROS source, the inner mitochondrial membrane [9,62]. Other factors contributing to mtDNA’s susceptibility include the absence of protective histone proteins, limited mtDNA repair activity, retention of superoxide anions within mitochondria, and the negatively charged environment of the mitochondrial matrix [9,59,63,64]. ROS can oxidize bases and the sugar-phosphate backbone of mtDNA or incorporate oxidized bases during DNA polymerization [26,65]. If incompletely repaired, DNA nicks and abasic sites accumulate, blocking mtDNA transcription and replication [26].

Susceptibility to oxidative damage follows the base reactivity order G > C > T >> A, with guanine being the most prone to oxidation, leading to 8-oxo-guanine formation and subsequent G-A mismatches [66,67]. Oxidative-generated DNA modifications have been systematically reviewed elsewhere [64,68]. Hydroxyl radicals readily attack the 2′-deoxyribose, creating carbon-centered sugar radicals that lead to strand incision and unique products (base propenal, 3′-phosphoglycolate, 5′-aldehyde) [64]. Although purine 8,5′-cyclo-2′-deoxyribonucleosides can form via sugar-base radical addition, they have not yet been detected in mtDNA [64]. On pyrimidines, hydroxyl radicals add at C5 or C6 to yield C_5_–OH–C_6_ or C_6_–OH–C_5_ pyrimidine radicals and downstream redox products [64,68]. The principal purine modification is C8 addition, producing 2,6-diamino-4-hydroxy-5-formamidopyrimidine/4,6-diamino-5-formamidopyrimidine or 8-hydroxy-guanine/8-oxo-7,8-dihydroguanine/8-oxo-7,8-dihydroadenine through different redox pathways [64,69]. Moreover, oxidative stress compromises mitochondrial integrity, promoting the release of modified mtDNA into the cytosol [67].

In most cardiovascular diseases, elevated ROS levels increase mtDNA damage and reduce oxidative phosphorylation [21,31,70,71]. Both clinical and animal studies demonstrate a tight link between mtDNA lesions and mitochondrial dysfunction in myocardial remodeling and heart failure [8,9,63]. Reports indicate oxidative mtDNA damage rises by over 50% in failing hearts [10]. The common mtDNA^4977^ deletion, a hallmark oxidative lesion, accumulates with age in human hearts and during vascular aging and atherosclerosis [72,73]. In atherosclerosis, conditions such as hyperglycaemia and smoking may further promote mtDNA damage via ROS [74,75], and such damage compromises metabolism, fostering endothelial dysfunction and pro-atherogenic phenotypic changes in vascular smooth muscle cells (VSMC) [11]. Notably, leukocyte levels of 8-hydroxy-2′-deoxyguanosine, an oxidative purine product, predict coronary artery disease risk in type 2 diabetes [69]. As the most common macrovascular complication of diabetes, atherosclerosis also involves ROS-mediated DNA strand breaks [76], with damage accumulating from early lesions and escalating with disease progression [77].

Beyond oxidative insults, non-oxidative mtDNA damage also contributes to cardiovascular diseases: heart failure arises from heart-specific mutant uracil-DNA glycosylase 1 expression [78], and atherosclerosis can result from ROS-independent mtDNA damage in smooth muscle cells and monocytes [75]. Additionally, cardiac hypertrophy, fibrosis, and failure occur following primary mtDNA damage induced by homozygous POLG mutations or prolonged zidovudine treatment [79,80].

### 3.2. MtDNA Mutations

Mutations in the mitochondrial genome give rise to a wide spectrum of diseases, the majority of which are maternally inherited and involve defects in oxidative energy metabolism [81,82]. MtDNA mutation rate significantly exceeds that of nDNA owing to increased oxidative damage, frequent replication errors, inefficient repair mechanisms, and absence of histone protection [83,84]. In adults, mtDNA mutations account for approximately 70% of mitochondrial diseases [85]. Depending on their location, these mutations may impair specific respiratory-chain proteins or disrupt overall mitochondrial protein synthesis if affecting tRNA or rRNA genes [86].

MtDNA mutations are broadly classified as point mutations or large-scale rearrangements [86]. Most point mutations occur in tRNA genes, with tRNA^LeuUUR^ and tRNA^Ile^ representing “hotspots” for cardiomyopathy-associated mtDNA variants [7]. Rearrangements are typically heteroplasmic, coexisting with varying proportions of wild-type genomes [7]. Under normal conditions, pathogenic mtDNA variants constitute less than 1% of the total mtDNA pool, rendering tissues effectively homoplasmic for the wild-type sequence [85,86]. MtDNA mutation only manifests clinically once its heteroplasmic load exceeds a tissue-specific threshold-often above 60%—which varies according to the reliance of each tissue on oxidative metabolism [7,85,87]. Consequently, the severity of cellular or tissue dysfunction and the overall phenotypic expression of mtDNA disease are largely determined by the mutant-to-wild-type genome ratio [7,85,88].

MtDNA mutations are strongly associated with cardiovascular disease [7], with specific examples including mtDNA abnormalities in dilated cardiomyopathy [89], increased mtDNA deletions in atrial fibrillation [31], the *MTTL1* m.3243A>G mutation and sudden cardiac death [90], and small deletions or heteroplasmic length variants in chronic atrial fibrillation [32]. Notably, approximately 20% of known pathogenic mtDNA point mutations are linked to cardiomyopathy, with tRNA mutations, particularly in tRNA^LeuUUR^ and tRNA^Ile^-constituting recognized hotspots [7]. Pathogenic variants in tRNA^Ile^ (m.4277T>C, m.4295A>G, m.4300A>G, m.4320C>T) are associated with mitochondrial hypertrophic cardiomyopathy, while m.4269A>G causes encephalocardiomyopathy and m.4317A>G leads to fatal infantile cardiomyopathy [91]. The m.3243A>G mutation in tRNA^LeuUUR^ commonly induces MELAS syndrome, frequently involving cardiac manifestations such as hypertrophic cardiomyopathy or Wolff-Parkinson-White syndrome [7,92]. Additionally, transgenic models support a causal role: accelerated accumulation of mtDNA mutations in mice recapitulates human disease phenotypes, including premature aging, dilated cardiac hypertrophy, and fatal congestive heart failure [93]. Kearns–Sayre syndrome—caused by mtDNA rearrangements—demonstrates how impaired energy supply disrupts high-demand tissues [7,94].

Together, these findings highlight the central role of mtDNA mutations-especially those affecting tRNA genes-in the pathogenesis of mitochondrial cardiomyopathies.

### 3.3. MtDNA Copy-Number Variations

MtDNA copy-number varies among individuals and even among different tissues within the same individual [95], serving as a candidate biomarker of mitochondrial function that reflects both mitochondrial abundance per cell and genome copies per mitochondrion [63,96]. MtDNA depletion, defined as a reduction to less than 30% of normal levels [91], arises from impaired biogenesis or excessive loss due to enzyme deficiencies [97], defective nucleotide metabolism [98], loss of mitochondrial fusion [99], drug toxicity [100,101,102], and other causes. Depletion disrupts the mitochondrial network, reduces cristae membrane content, and leads to a circular morphology of the remaining cristae [103].

Conversely, in response to mitochondrial dysfunction, nuclear-encoded factors may drive disproportionate mitochondrial biogenesis, leading to over-replication of the mtDNA and mitochondrial proliferation [95]. Accordingly, increased mtDNA copy-number in mitochondrial diseases may represent a compensatory mechanism to alleviate bioenergetic deficits, delay disease onset, and modulate phenotypic severity [95,104]. This compensatory increase helps explain why patients harboring high loads of deletion mutants sometimes present with only mild mitochondrial myopathy [95].

In humans, mtDNA content correlates positively with mitochondrial gene transcription and serves as an indicator of mitochondrial dysfunction [80]. Reduced mtDNA copy-number underlies severe cardiac disorders such as cardiomyopathy [91], pathological cardiac remodeling [105,106], atrial fibrillation [107], and heart failure [10,108]. Certain chemotherapeutic agents-including doxorubicin [100,101,102] and nucleoside reverse transcriptase inhibitors [109,110]—exert cardiotoxicity by inhibiting DNA replication and depleting mtDNA. In Chagasic cardiomyopathy, poly (ADP-ribose) polymerase 1 translocates to mitochondria and binds mitochondrial POLγ, impairing mtDNA maintenance and exacerbating mitochondrial dysfunction, oxidative stress, and cardiac remodeling [105]. A case of severe dilated mitochondrial cardiomyopathy was linked to a novel homoplasmic tRNA^Ile^ mutation accompanied by profound mtDNA depletion [91]. Additionally, patients with heart failure of diverse etiologies exhibit reduced mtDNA content alongside decreased levels of mtDNA-encoded proteins [111].

### 3.4. MtDNA Methylation

Epigenetic mechanisms produce heritable phenotypic changes via chromosome modifications rather than alterations in DNA sequence, enabling adaptive responses to environmental stimuli [40,112]. Among these, DNA methylation is the most extensively studied modification and plays a pivotal role in establishing cellular identity. Dysregulated mtDNA methylation has been implicated in aging and various diseases [41,113,114]. MtDNA methylation involves transfer of a methyl group from S-adenosyl methionine to DNA bases, with primarily adenine in the D-loop of the light (L) strand [4,114,115]. Increased methylation in this region generally suppresses transcription [4], thereby affecting mitochondrial quality control and often triggering the integrated stress response [116]. DNA methyltransferases (DNMTs) mediate mtDNA methylation, with DNMT1 being the most abundant and active form throughout adulthood [116]. DNMT1 harbors a mitochondrial targeting sequence, which directs its import into mitochondria, where it binds mtDNA and catalyzes D-loop methylation [117,118]. Upregulation of mitochondrial DNMT1 exerts gene-specific effects: it represses ND6 (the sole protein-coding gene on the L strand) while enhancing ND1 expression on the H strand [41]. Both ND1 and ND6 encode subunits of complex I critical for cardiac contractility [119]. Recent advances in detection methods have identified N^6^-methyldeoxyadenine (6mA)—a methylation mark once thought restricted to bacteria—in eukaryotes [120]. Its regulatory role in mitochondrial stress has been reported [120,121]. Methyltransferase-like protein 4 (METTL4), a putative mammalian methyltransferase, catalyzes mtDNA 6mA methylation, leading to repression of transcription and reduction in mtDNA copy-number [121].

Although the prevalence and functional significance of mtDNA methylation remain debated, numerous studies have documented correlations between mtDNA methylation (and hydroxymethylation) patterns and cell type, differentiation state, age, and disease status [4,41]. In metabolically active organs such as the heart, altered mtDNA methylation impacts bioenergetics and can promote systemic inflammation, vascular endothelial dysfunction, and myocardial injury [116]. The methylation and downregulation of mitochondrial gene *COX2,* which encodes COII, are biomarkers of aging in heart mesenchymal stem cells [122]. Moreover, in platelets from patients with cardiovascular disease, increased methylation of mtDNA encoding cytochrome c oxidase and tRNA leucine 1 has been reported [123]. Mitochondrial-targeted overexpression of DNMT1 also impairs mitochondrial gene expression, compromises mitochondrial function, and reduces VSMC contractility [118]. Finally, mtDNA methylation appears to influence clinical presentation in coronary artery disease: patients with acute coronary syndrome exhibit higher mtDNA methylation and lower mtDNA copy-number compared to those with stable coronary artery disease [124].

### 3.5. MtDNA Release

Under stress, mitochondria can release mtDNA into the cytosol, where it acts as a damage-associated molecular pattern (DAMP) and triggers inflammation [52,125]. During apoptosis, pro-apoptotic proteins BAX and BAK promote mitochondrial outer membrane permeabilization (MOMP), allowing release of intermembrane-space proteins—such as cytochrome c—and subsequent activation of caspases, culminating in cell death [126,127]. However, cell death can also proceed via caspase-independent pathways, which are accompanied by mtDNA leakage [126]. When caspases are inhibited, BAX/BAK-mediated MOMP drives herniation of the inner mitochondrial membrane into the cytosol, carrying matrix components, including mtDNA [126,128,129]. Rather than preventing herniation, apoptotic caspases function to clear dying cells and suppress mtDNA-induced innate immune signaling via the cGAS–STING pathway, thereby maintaining immunological silence during apoptosis [129].

Large mitochondrial membrane pores can also form independently of BAX/BAK. Under oxidative stress, voltage-dependent anion channel 1 (VDAC1)—the most abundant outer-membrane protein, which regulates Ca^2+^ influx, metabolism, inflammasome activation, and cell death—oligomerizes to create large pores that facilitate mtDNA release [130]. It has been proposed that VDAC1 oligomers mediate mtDNA efflux under moderate stress, whereas BAX/BAK macropores predominate during severe stress or apoptosis [130]. Another debated mechanism involves the mitochondrial permeability transition pore (MPTP), a nonspecific 2–3 nm inner-membrane channel formed in response to Ca^2+^ overload and other stimuli [66,131]. The MPTP permits passage of molecules smaller than ~1500 Da and may allow egress of mtDNA fragments [66], although its precise role remains controversial and is sometimes conflated with VDAC1 function [131,132].

Mitochondria-derived vesicles have also been suggested as carriers of mtDNA; these small vesicles bud from mitochondria carrying proteins and lipids while preserving membrane integrity [131,133]. However, evidence for mtDNA incorporation into mitochondria-derived vesicles and its subsequent release into the cytosol to activate cGAS remains limited [131]. Finally, excessive mitochondrial fission can disrupt membrane integrity, causing rupture and release of mtDNA into the cytosol [18].

Myocardial mitochondrial dysfunction and consequent mtDNA release are crucial pathophysiological hallmarks or mediating factors for the pathogenesis of myocarditis [134], post-MI cardiac dysfunction [18], myocardial ischemia/reperfusion injury (MI/RI) [23], and thrombosis [135]. Perfusion of the coronary circulation with mtDNA fragments during heart ischemia increases infarct size, underscoring mtDNA’s cardiotoxic potential [26]. In an atherosclerosis mouse model, oxidized mtDNA is released via VDAC1-dependent MPTP opening and activates the STING–PERK pathway [132]. Similar interactions between released mtDNA and inflammation have been observed in animal models of hypertension [15] and doxorubicin-induced cardiotoxicity [136]. Circulating mtDNA has also been proposed as a blood-based biomarker for atrial fibrillation [137].

The lipotoxicity-mediated mtDNA release has also been reported. Upon cellular entry, fatty acids may be esterified into triglycerides for storage in lipid droplets [138,139]. Upon demand, stored triacylglycerols are released and hydrolyzed into fatty acids, which are subsequently converted to fatty acyl-CoA [139]. While fatty acyl-CoA normally undergoes β-oxidation in mitochondria to support energy production, cardiac cells may also divert it toward the synthesis of harmful lipids such as ceramides, diacylglycerols, and triglycerides [140]. The resulting lipid accumulation causes lipotoxicity, promoting endoplasmic reticulum stress, apoptosis, and inflammation [138]. Importantly, lipotoxicity can trigger mtDNA release, activating cGAS–STING signaling and inflammatory pathways, ultimately leading to cardiac cell death and fibrosis [29,30].

### 3.6. Autophagy

Autophagy serves as another line of defense when insults overwhelm the protective antioxidant, repair pathways, and inhibition of mtDNA release. This homeostatic mechanism sequesters harmful cytosolic components within double-membrane autophagosomes, which subsequently fuse with lysosomes to degrade their contents [125,141]. Under low-energy conditions, AMP-activated protein kinase (AMPK) is activated and phosphorylates ULK1 (unc-51–like kinase 1), initiating phagophore formation. This curved double membrane structure expands via autophagy-related (ATG) proteins into a mature autophagosome [127]. Microtubule-associated protein 1 light chain 3 beta (LC3B), a member of the ATG8 family, integrates into the growing phagophore and drives membrane elongation, of which LC3B-I turns into lipidated form termed LC3B-II that binds to the phagophore [127].

Mitophagy, the selective removal of damaged mitochondria, begins when PTEN-induced kinase 1 (PINK1) accumulates on the outer mitochondrial membrane and recruits Parkin. Parkin ubiquitinates mitochondrial surface proteins, facilitating recruitment of LC3-binding adaptors and targeting mitochondria for autophagic degradation [142]. Upon mitochondrial stress, PINK1 directly phosphorylates and activates Parkin on Ser65 in its ubiquitin-like domain, which leads to ubiquitination of substrates on damaged mitochondria that function as autophagy-mediated degradation signals [143]. Mitophagy mediated by this signaling pathway attenuates pathological cardiac alternation via inhibiting the mtDNA release [144,145]. On the other hand, although PINK1 and Parkin are not encoded by mitochondrial genes, the 3733G>C mutation in mtDNA downregulated PINK1/Parkin-mediated mitophagy pathway [146]. In selective autophagy, autophagy adaptors act as selective takers, which contain both a ubiquitin-binding domain for recognizing ubiquitin chains and an LC3-interacting region for recruiting phagophore membranes [143]. During mitophagy, all known autophagy adaptors are recruited to damaged mitochondria in a PINK1/Parkin-dependent manner [147].

In addition to mitochondrial proteins functioning as substrates of PINK1/Parkin-mediated ubiquitination, mitophagy also relies on various mitochondrial cargo receptors, including BCL2-interacting protein 3 (BNIP3), BNIP3-like protein (BNIP3L, known as NIX), FUN14 domain containing 1(FUNDC1), and others [148,149]. BNIP3 and NIX function primarily under stress, which integrate into the outer mitochondrial membrane via their carboxy terminal transmembrane domains, with the remaining bulk of both proteins facing into the cytosol, where a conserved LC3 interaction region near the amino-terminal end interacts with processed LC3/GABARAP [148]. NIX undergoes ubiquitylation by Parkin to facilitate recruitment of the autophagy adaptor NBR1, which promotes formation of autophagosomes surrounding mitochondria by simultaneously binding to ubiquitin and LC3/GABARAP, while BNIP3 interacts with and stabilizes PINK1 on the outer mitochondrial membrane, leading to Parkin translocation to mitochondria [143]. Cardiac progenitor cells harboring mtDNA mutations fail to efficiently activate the mitophagy in response to differentiation, while this adverse situation is fortunately improved by NIX-and FUNDC1-mediated mitophagy [149].

Autophagosomes carrying impaired mitochondria fuse with the lysosomes and deoxyribonuclease (DNase) degrades mtDNA in the autophagy system to eliminate damaged mtDNA [141,150,151]. However, mtDNA that escapes autophagic clearance elicits inflammatory responses mediated by toll-like receptor (TLR) 9, the NLRP3 inflammasome, or the cGAS-STING pathway [16,125,141,152]. TLR9 recognizes unmethylated CpG motifs in mtDNA delivered to lysosomes by autophagosomes [153,154]; intriguingly, TLR9 activation via this route also contributes to mitochondrial biogenesis, suggesting context-dependent roles based on the source of the TLR9 ligand and how or where the receptor/ligand interaction takes place [154]. Meanwhile, autophagy proteins regulate innate immune responses by limiting NLRP3 inflammasome-mediated mtDNA release [125].

Apoptosis-induced autophagy can occur independently of caspase activation: BAX/BAK-mediated mitochondrial permeabilization rapidly activates AMPK and ULK1, driving autophagic flux that degrades mitochondria and reduces mtDNA release and downstream innate immune signaling [127]. However, it is suspected that cGAS/STING activation occurs too quickly for autophagic clearance to inhibit this early step effectively. For this question, the authors speculated that BAX/BAK-stimulated autophagy causes the cytokine to be sequestered rather than released, but it remains to be determined [127]. Moreover, STING itself can induce autophagy from the ER-Golgi intermediate compartment, as evidenced by LC3 lipidation and autophagosome formation, although the precise mechanism remains unclear [128]. We hypothesize that STING activation at the ER-Golgi intermediate compartment stimulates the local production of autophagy-specific phosphoinositides or facilitates the transfer of lipids to create a lipid microenvironment conducive to the elongation and curvature of phagophores. Because phosphatidylinositol 4-phosphate, the most abundant cellular phosphoinositide with the highest abundance at the trans-Golgi network, is an energy source for lipid-transfer [155]. Lipid transfer plays a critical role in membrane remodeling and is directly implicated in autophagy [156].

Physiologically, mitophagy maintains mitochondrial quality by removing damaged organelles. In some metabolic disorders and cardiovascular diseases, mitophagy is often impaired, leading to the accumulation of dysfunctional mitochondria, elevated mtDNA fragments, and chronic inflammation [157]. For instance, mtDNA, if escaping from autophagy, can provoke inflammation in cardiomyocytes, even myocarditis, and dilated cardiomyopathy [141]. In atherosclerotic plaques, endogenous mtDNA complexed with LL-37 evades DNase II and autophagic degradation, persistently activating TLR9 and driving chemokine and cytokine production that exacerbates lesion development [16]. Similarly, impaired mitophagy facilitates NLRP3 inflammasome activation in atherosclerosis [158], and in silica nanoparticle exposure-induced cardiac adverse events, defective autophagic flux permits cytosolic mtDNA accumulation and cGAS-STING-mediated pyroptosis [152]. During myocardial ischemia/reperfusion or hypoxia/reoxygenation, autophagy deficiency promotes mitochondrial fission, mtDNA release, and sterile inflammation, i.e., the activation of the STING/IRF3 axis [24].

### 3.7. Inflammation

Because mtDNA contains unmethylated CpG motifs similar to bacterial DNA, it elicits stronger inflammatory responses than nDNA [16]. When mtDNA enters the cytosol, extracellular space, or circulation, it activates multiple pattern-recognition receptors, triggering pro-inflammatory and type I interferon responses [2]. Different mtDNA topology affects different inflammatory responses of cardiomyocytes [159]. As a prototypical DAMP, mtDNA–driven sterile inflammation plays a central role in the pathogenesis of diverse cardiovascular diseases [160,161].

Endolysosomal TLR9 recognizes unmethylated CpG motifs in mtDNA released from injured cells. Upon binding mtDNA, TLR9 initiates a signaling cascade that activates nuclear factor-κB (NFκB) and induces expression of pro-inflammatory cytokines such as TNF-α, IL-1β, and IL-6, as well as assembly of the NLRP3 inflammasome [125,141,160,162,163,164,165]. In the injured heart, these cytokines elevate intracellular ROS, which further compromise mitochondrial membrane integrity and exacerbate cell death [134]. NLRP3 activation itself appears to depend on hexokinase dissociation from mitochondria, thereby promoting the oligomerization of voltage-dependent anion channel (VDAC) on the outer membrane; together with cytosolic mtDNA fragments, oligomerized VDAC recruits NLRP3 to the mitochondrial surface for inflammasome assembly [166]. Oxidized mtDNA released during apoptosis can also directly bind and activate NLRP3 in primed macrophages, driving IL-1β secretion [167].

The cGAS-STING signaling pathway senses pathogenic DNA and initiates a tightly regulated signaling cascade to trigger various inflammatory effectors [30,168]. Cytosolic mtDNA is sensed by cGAS, which catalyzes synthesis of 2′,3′-cyclic GMP-AMP (2′,3′-cGAMP) from ATP and GTP upon DNA binding [128,169]. cGAMP then functions as a secondary messenger activating STING, leading to phosphorylation of interferon regulatory factor 3 (IRF3) and NFκB via the TANK-binding kinase 1 (TBK1) and I-kappa-B kinase (IKK), respectively [30,128,168]. Phosphorylated IRF3 dimerizes and translocates to the nucleus to drive type I interferon transcription [33], while NFκB initiates the transcription of pro-inflammatory cytokines implicated in atherosclerosis, pulmonary arterial hypertension, and myocarditis [132,134,170]. In addition, it may also be involved in metabolic regulation by rewiring the metabolic circuitry, such as ornithine decarboxylase-putrescine metabolic flux [171]. Beyond these canonical pathways, STING can also activate PERK in endothelial cells [132], engage MAPKs, stimulate NLRP3, trigger autophagy, and even induce lytic cell death, underscoring its multifunctional role in inflammation and cell fate [128,157,168].

Clinically, robust myocardial infarction (MI)-mediated inflammation contributes to adverse remodeling and dysfunction across cardiovascular contexts. In this process, cGAS-STING-driven macrophage polarization toward an M1 phenotype exacerbates tissue damage and remodeling; conversely, inhibiting this axis enhances wound healing, neovascularization, and survival post-MI [19,20]. Additionally, NFκB activated by the STING, transactivates ornithine decarboxylase-putrescine metabolic flux to prompt the pathogenesis of cardiac hypertrophy [171]. In hypertension, elevated circulating mtDNA coupled with impaired DNase activity activates TLR9, raising arterial pressure and promoting vascular dysfunction [172]. In atherogenesis, ROS-induced mtDNA damage and release fuel vascular inflammation and VSMC phenotypic switching [17,77], while VCAM1-mediated aberrant mtDNA synthesis in macrophages triggers STING-dependent inflammation and lesion progression [173]. Similarly, STING signaling drives VSMC senescence and fibrous-cap thinning in chronic-kidney-disease-associated atherosclerosis [174]. Finally, activated NLRP3 in cardiomyocytes promotes collagen deposition, left ventricular hypertrophy, fibrosis, and diastolic dysfunction [30].

### 3.8. Pyroptosis

Pyroptosis is an inflammatory form of programmed cell death characterized by caspase-1-dependent formation of membrane pores, resulting in the release of pro-inflammatory cytokines IL-1β and IL-18 [21]. This process is closely linked to activation of the NLRP3 inflammasome, a multimeric complex composed of NLRP3, the adaptor protein ASC (apoptosis-associated speck-like protein containing a CARD), and caspase-1 [21]. Upon assembly, the inflammasome cleaves pro–caspase-1 into active caspase-1, which in turn processes pro-IL-1β and pro–IL-18 into their mature forms and simultaneously cleaves gasdermin D (GSDMD) to generate the pore-forming N-terminal fragment (GSDMD-N). GSDMD-N mediates secretion of IL-1β and IL-18 and disrupts plasma membrane integrity, culminating in pyroptotic cell death [30,175].

In cardiovascular pathology, NLRP3 inflammasome activation and elevated IL-1β/IL-18 levels drive recruitment of macrophages to aortic lesions, promoting foam-cell formation and plaque progression [176]. Within the myocardium, NLRP3 inflammasome activation in cardiac fibroblasts stimulates IL-1β release and cardiomyocyte pyroptosis, exacerbating cardiac inflammation and adverse remodeling [158]. Additionally, this inflammatory cascade and pyroptosis have been observed in myocardial ischemia [21].

Metabolic stress further accentuates pyroptosis in the heart. In diabetic cardiomyopathy, free fatty acid–induced mtDNA release into the cytosol activates cGAS–STING, triggering pyroptosis and inflammatory responses that drive cardiac hypertrophy [30]. Similarly, exposure to silica nanoparticles disrupts autophagy, leading to cytosolic mtDNA accumulation and cGAS–STING–dependent pyroptosis, as described above [152]. Intriguingly, recent data suggest that cGAS–STING–NLRP3–mediated pyroptosis may enhance the efficacy of radiotherapy in hypertrophic cardiomyopathy, highlighting context-dependent roles for this pathway [177].

**Table 1 antioxidants-14-01138-t001:** Representative examples of cardiovascular pathologies driven by mtDNA abnormalities.

Clinical Condition	Mechanism	Evidence/Example	Reference(s)
Hypertension	MtDNA mutations; released mtDNA activates innate immune responses, contributing to vascular dysfunction	m.A14696G and m.A14693G mutations lead to failure in tRNA^Glu^ metabolism	[12]
		m.15992A>G and m.15077G>A mutations may disturb mitochondrial function	[13]
		m.T10410C and m.T10454C mutations affect the structure and function of tRNA^Arg^	[14]
		Released mtDNA mediates inflammation	[15]
		Elevated circulating mtDNA and impaired DNase activity activate TLR9	[172]
Atherosclerosis	Damaged mtDNA escapes autophagy and promotes vascular inflammation and VSMC senescence	MtDNA promotes atherosclerosis through mitochondrial dysfunction	[71]
		MtDNA^4977^ deletion is a hallmark oxidative lesion, accumulating in vascular aging and atherosclerosis	[72,73]
		Atherosclerosis can result from ROS-independent mtDNA damage in smooth muscle cells and monocytes	[75]
		Oxidized mtDNA is released and activates the STING–PERK pathway	[132]
		MtDNA complexed with LL-37 evades autophagic degradation	[16]
		Impaired mitophagy facilitates NLRP3 inflammasome activation	[158]
		ROS-induced mtDNA damage and release promote vascular inflammation and VSMC phenotypic switching	[11,17,77]
		Aberrant mtDNA synthesis in macrophages exacerbates STING-dependent inflammation and atherosclerosis	[173]
		STING signaling drives VSMC senescence and fibrous-cap thinning	[174]
		NLRP3 inflammasome activation and elevated IL-1β/IL-18 levels drive recruitment of macrophages to aortic lesions, promoting foam-cell formation and plaque progression	[176]
		MtDNA repair can regulate NLRP3 inflammasome and prevent atherosclerosis	[178]
Myocardial infarction	Damaged mtDNA is released and triggers severe inflammatory responses, exacerbating tissue damage	Myocardial mitochondrial impairment, mtDNA release, and subsequently STING/p65 activation mediate post-MI cardiac dysfunction	[18]
		cGAS functions as a cytosolic DNA receptor, promoting macrophage polarization and governing myocardial ischemic injury	[19,20]
		PCSK9 initiates mtDNA damage, and induces pyroptosis	[21]
		Protection of mitochondria and mtDNA can ameliorate isoproterenol-induced MI	[22]
Myocardial ischemia/reperfusion injury	DNA methylation, mtDNA damage, and release amplify sterile inflammation	Age-associated DNA methylation augments cardiac sensitivity towards MI/RI	[119]
		Increased mtDNA release enhances pro-inflammatory cytokines	[23]
		Autophagy deficiency promotes mitochondrial fission, mtDNA release, and sterile inflammation	[24]
		ROS can initiate DNA single-strand breakage and severe lesions subsequently	[25]
		Perfusion of coronary circulation with free mtDNA fragments aggravates infarct	[26]
Heart Failure	MtDNA lesions, mutations, and depletion impairing OXPHOS and ATP production	Oxidative mtDNA damage mediates cardiac dysfunction	[9]
		ROS increase can lead to a catastrophic cycle of mtDNA damage and cellular injury in heart failure	[63]
		MtDNA toxicity can impair dynamics and function of mitochondria, leading to heart failure	[78]
		Angiotensin II-mediated mtDNA oxidative damage, homozygous POLG mutations, or prolonged zidovudine treatment contribute to cardiac hypertrophy, fibrosis, and failure	[79]
		Accumulation of mtDNA mutations induces premature aging, dilated cardiac hypertrophy, and fatal congestive heart failure	[93]
		MtDNA copy number depletion is an independent risk factor for heart failure	[108]
		Reduced mtDNA replication and depletion of mtDNA impair mitochondrial biogenesis	[10]
		Patients with heart failure exhibit reduced mtDNA content and mtDNA-encoded proteins	[111]
		TFAM overexpression ameliorates mitochondrial deficiencies, increases mtDNA copy-number, and improves cardiac failure	[27,28,179]
Diabetic Cardiomyopathy	Hyperglycemia induces mtDNA oxidative damage and release, activating cGAS-STING-mediated pyroptosis	Leukocyte levels of 8-hydroxy-2′-deoxyguanosine predict coronary artery disease risk in type 2 diabetes	[69]
		Oxidized mtDNA release and triggering cGAS-STING-mediated pyroptosis	[30]
		Diabetic cardiomyopathy reduces mtDNA replication and transcription, together with impairing mitochondrial ultrastructure	[180]
		Greater mtDNA damage is found in patients with diabetes mellitus and clinical atherosclerosis	[76]
Cardiac Hypertrophy	MtDNA mutations, mtDNA damage and chronic inflammation promote fibrosis and hypertrophy	Mutations in tRNA^Ile^ including m.4277T>C, m.4295A>G, m.4300A>G, m.4320C>T	[91]
		Oxidative stress-derived mtDNA damage and deletion are partly linked to cardiac hypertrophy	[72]
		MtDNA lesions cause a vicious cycle with decreased cardiac bioenergetics and ROS accumulation	[80]
		NFκB activated by the STING, prompts cardiac hypertrophy	[171]
Dilated cardiomyopathy	MtDNA mutations and impaired autophagic flux	Pathological mtDNA mutations leading to abnormal mitochondria	[89]
		A homoplasmic tRNA^Ile^ mutation accompanied by profound mtDNA depletion	[91]
		MtDNA that escapes from autophagy provokes myocarditis and dilated cardiomyopathy	[141]
Arrhythmia	MtDNA oxidative lesions, mutations, and copy-number variations	Oxidative lesions and mtDNA deletions in cardiomyocytes are increased in atrial fibrillation	[31]
		Kearns–Sayre syndrome, caused by large-scale mtDNA rearrangements, leads to progressive conduction system degeneration.	[7,94]
		MtDNA mutations are detected in patients with chronic atrial fibrillation	[32]
		MtDNA copy-number is a risk factor for atrial fibrillation	[107]
Others	MtDNA mutations	m.3243A>G mutation of MTTL1 in sudden cardiac death	[90]
		m.4269A>G mutation causes encephalocardiomyopathy and m.4317A>G leads to fatal infantile cardiomyopathy	[91]
		m.3243A>G mutation in tRNA^LeuUUR^ commonly produces MELAS syndrome	[7,92]
	MtDNA copy-number variations	In Chagasic cardiomyopathy, poly (ADP-ribose) polymerase 1 impairs mtDNA maintenance	[105]
	MtDNA methylation abnormality	The methylation and downregulation of COX2 are biomarkers of aging in heart mesenchymal stem cells	[122]
		Altered mtDNA methylation can promote systemic inflammation, vascular endothelial dysfunction, and myocardial injury	[116]
		Overexpression of DNMT1 impairs mitochondrial gene expression, compromises mitochondrial function, and reduces VSMC contractility	[118]
	MtDNA release	Impaired mitochondria and released mtDNA mediate myocarditis	[134]
		Released mtDNA induces platelet activation, leading to thrombosis	[135]
	Impaired autophagy, mtDNA release, and pyroptosis	In cardiomyocytes exposed to silica nanoparticles, defective autophagic flux permits cytosolic mtDNA accumulation and cGAS-STING-mediated pyroptosis	[152]
	Pyroptosis	NLRP3 activation in cardiac fibroblasts stimulates cardiomyocyte pyroptosis, exacerbating cardiac inflammation and adverse remodeling	[158]

## 4. Therapeutic Strategies Targeting mtDNA and Its Related Pathways in Cardiovascular Diseases

Maintaining mtDNA integrity and function, and interrupting its downstream inflammatory and cell-death cascades, are crucial for preserving mitochondrial health and preventing cardiovascular disease. Table 2 summarizes current mtDNA-targeted interventions and their mechanisms.

### 4.1. Repairs of mtDNA Oxidative Damage

The circular structure and proximity of mtDNA to the respiratory chain render it highly susceptible to ROS-mediated damage. In the context of MI/RI, a burst of ROS leads to extensive mtDNA lesions that contribute to cardiomyocyte dysfunction and death [9,11,25,63,181]. Ecklonia cava extract [182] and oleoylethanolamide [183] have been shown to mitigate mtDNA oxidative damage: the former attenuates vascular calcification in hypertensive models, while the latter counteracts STING1-accelerated VSMC senescence-associated vascular calcification. In heterozygous mitochondrial superoxide dismutase knockout mice, overexpression of the mitochondrial helicase Twinkle enhances repair of oxidative mtDNA lesions and protects against cardiomyopathy [184]. Additionally, in spontaneously hypertensive rats, miRNA-21 was found to upregulate translation of mtDNA-encoded cytochrome b, reducing ROS production and ameliorating hypertension-related cardiac pathology [185].

ROS attack both bases and the sugar-phosphate backbone of mtDNA, resulting in mutations, deletions, and strand breaks [26,66]. To counteract these insults, mitochondria rely on a robust base-excision repair (BER) pathway that mitigates oxidative and other minor base damage. BER is initiated by lesion-specific DNA glycosylases possessing apurinic/apyrimidinic (AP) lyase activity, which recognize and remove oxidized or otherwise aberrant bases [2,26,181,186]. The resultant abasic site is processed by AP endonucleases and phosphodiesterases to remove the deoxyribose moiety; POLγ then fills the gap, and DNA ligase III seals the nick [181,186]. Guanine is particularly prone to oxidation, forming 8-oxo-guanine, which is excised by the glycosylase OGG1 [66,181,187]. Enhanced OGG1 activity confers significant protection across multiple cardiac injury models, including MI/RI [26], diabetic cardiomyopathy [186], atherogenesis [178], and cardiac failure [188].

In addition to OGG1, endonuclease III (Endo III) is another mitochondria-targeted DNA glycosylase/AP lyase peptide that preferentially targets oxidized pyrimidines. This fusion protein, Exscien1-III, provided pronounced cardioprotection in mouse models of MI/RI and transverse aortic constriction–induced heart failure [26,189]. However, some studies suggest that BER alone may not suffice to preserve cardiac function in vivo following MI/RI, potentially due to limitations in functional assessment techniques, observation time windows, or the overwhelming ROS burden that exceeds repair capacity [26,181,190,191]. To address this, combined therapy with Endo III and DNase I has been employed: Endo III minimizes DAMP generation by repairing mtDNA lesions, while DNase I degrades irreversibly damaged mtDNA fragments that escape repair [26]. Therefore, such combination therapy targeting both mtDNA damage and its immunostimulatory properties, alongside other therapeutic strategies, is expected to exhibit broad applicability in cardiovascular protection. For instance, Ecklonia cava extract [182], RU.521 [192], and Hirudin [193] collectively ameliorate mitochondrial dysfunction-induced hypertension or hypertensive cardiac injury by targeting multiple stages of the pathological cascade: inhibiting mtDNA oxidative damage, suppressing cGAS activation, and blocking NLRP3 inflammasome assembly, respectively.

### 4.2. MtDNA Quality Control

Given the central role of TFAM in maintaining mtDNA integrity, therapies that restore TFAM levels confer cardioprotection. In MI/RI rat models, lycopene treatment [194] and the traditional Chinese formulation Huoxue Huatan Decoction [195] preserve cardiomyocyte viability by upregulating TFAM and reducing mtDNA oxidative damage. In angiotensin II-induced hypertensive mice, voluntary exercise enhances endothelial mitochondrial remodeling and lowers blood pressure by promoting p53–TFAM binding, thereby improving mtDNA integrity [196]. Diabetic hearts exhibit diminished TFAM expression and activity, correlated with impaired ETC function; these defects are reversed by resveratrol [54]. Moreover, TFAM overexpression [27,28,179] and TFAM acetylation [197] mitigate mtDNA depletion and mitochondrial dysfunction in models of MI, volume overload, and transverse aortic constriction-induced heart failure. Carvedilol—a third-generation, nonselective β-blocker used in heart failure-stimulates mitochondrial biogenesis via the PGC-1α/TFAM axis in human umbilical vein endothelial cells [198]. In cardiomyocyte hypertrophy models, recombinant human TFAM protein increases mtDNA copy number and inhibits nuclear factor of activated T cells, exerting further cardioprotective effects [199].

Beyond TFAM, mitochondrial quality control encompasses biogenesis, dynamics, and autophagic flux, all of which can be modulated pharmacologically. Agents such as Tat-Beclin 1 [200], sappanone A [201], β-hydroxybutyrate [202], suberoylanilide hydroxamic acid [203], ablating lysocardiolipin acyltransferase 1 [204], astragaloside IV [205], resveratrol [206], and oleuropein [207] reduce mtDNA damage, elevate mtDNA copy number, and improve mitochondrial function in cardiovascular diseases. Activation of peroxisome proliferator–activated receptors (PPARs) also regulates mitochondrial homeostasis [208]: PPARα activation [209] and PPAR 1α deacetylation [210], protect against hyperlipidemia—induced vascular calcification and atherogenesis, respectively. Additionally, high-density lipoprotein (HDL) facilitates cholesterol efflux, maintains the integrity and functionality of mitochondria [211,212], and exhibits antioxidant, anti-inflammatory, and other beneficial properties, contributing to its cardiovascular protective effects [213,214]. The beneficial properties of HDL may interact positively with mtDNA, as suggested by the consistent levels observed between HDL (or HDL-C) and mtDNA [215,216]. Additional interventions—Dl-3-n-butylphthalide [217], the SIRT1 activator RT1720 [218], and Twinkle overexpression [219]—augment mtDNA content and biogenesis in H9c2 cardiomyocytes after H_2_O_2_ insult and in MI models. In age-related cardiac hypertrophy, enalapril confers protection by attenuating oxidative damage, increasing mitochondrial mass, and preserving quality-control mechanisms [72].

### 4.3. MtDNA Content Maintenance

As MtDNA copy-number correlates positively with mitochondrial gene transcription and reflects organelle functionality [80]. Calcitriol supplementation [220] and chronic aerobic exercise [221] increase mtDNA copy-number and improve mitochondrial function, ameliorating age-related hypertension and aortic sclerosis with endothelial dysfunction, respectively. Overexpression of Twinkle, the nuclear-encoded mtDNA helicase, enhances mtDNA replication, raising copy number and conferring protection in volume overload-induced [179] and pressure overload-induced [222] heart failure. Similarly, mitochondrial topoisomerase I relieves mtDNA topological stress and entanglements generated during replication and transcription; its activity shields against doxorubicin—induced cardiotoxicity by maintaining mtDNA topology [223]. Exercise training [180] and recombinant human glucagon-like peptide-1 [224] preserve mtDNA replication and transcription, prevent copy-number decline, and ameliorate mitochondrial ultrastructural defects in diabetic cardiomyopathy and intermittent hypoxia-induced cardiac injury. In addition, omega-3 fatty acids, having been reported as an inducer of mitochondrial biogenesis and function, and their appropriate supplementation exerts their neuro-cardio protective role [225].

Paradoxically, mtDNA–depleted J774A.1 murine macrophages—generated by ethidium bromide treatment—exhibit resistance to oxidized low-density-lipoprotein-induced pyroptosis [226]. This apparent contradiction likely reflects reduced release of mtDNA as a DAMP, thereby attenuating inflammatory cell death.

### 4.4. Regulation of mtDNA Methylation

DNMT1 is the principal enzyme catalyzing D-loop methylation of mtDNA. Inhibition of DNMT1 by 5-azacytidine attenuates mtDNA hypermethylation and confers protection against MI/RI [117,118,227]. One key target of mtDNA methylation is the COX2 gene, which encodes subunit II of cytochrome c oxidase (Complex IV). In human cardiac mesenchymal stem cells, age-associated senescence is accompanied by COX2 hypermethylation and downregulation; treatment with the DNA methyltransferase inhibitor 5-aza-2′-deoxycytidine restores COX2 expression and improves mitochondrial function [122].

Beyond cytosine methylation, excessive N^6^-methyladenine (6mA) in mtDNA—mediated by methyltransferase-like protein 4 (METTL4)—impairs mitochondrial gene expression and contributes to heart failure. Genetic or pharmacologic strategies that normalize METTL4 activity and reduce mtDNA 6mA levels alleviate cardiac dysfunction in experimental models [40]. However, developing therapies that selectively target mitochondrial epigenetics remains challenging due to systemic effects on nuclear DNA. For example, systemic administration of 5-aza-2′-deoxycytidine induces genomic hypomethylation, which can lead to genomic instability and tumorigenesis [4]. Tailored delivery systems or mitochondria-specific inhibitors will be essential to exploit mtDNA methylation as a therapeutic avenue without off-target genomic consequences [4].

### 4.5. Inhibition of mtDNA Release

Beyond repairing oxidative lesions, strategies that prevent mtDNA efflux from damaged mitochondria have shown therapeutic promise. Epigallocatechin-3-gallate attenuates mtDNA release under stress conditions, thereby reducing downstream inflammation [228]. Similarly, blockade of cell-surface nucleolin-using midkine or the aptamer AS1411 suppresses mtDNA uptake by cardiomyocytes during hypoxia-reoxygenation, mitigating inflammatory injury [229].

In the context of acute myocardial infarction, mitochondrial impairment and mtDNA release drive STING-p65 signaling and cardiomyocyte dysfunction [18]. Genetic deletion of large tumor suppressor kinase 2 (LATS2) prevents mtDNA liberation into the cytosol, thereby inhibiting STING–p65 activation and attenuating post-infarction injury [18]. Pharmacologic agents such as astaxanthin [230], thymoquinone [231], and octreotide [22] similarly reduce isoproterenol-induced myocardial damage by decreasing oxidative stress, preserving mitochondrial morphology and function, and restoring mtDNA copy-number.

Metabolic derangements also exacerbate mtDNA release: hyperglycemia impairs mitochondrial integrity via the SIRT1–AMPK–PGC1α axis, promoting secretion of mtDNA-enriched vesicles in diabetic cardiomyopathy. Both exogenous and endogenous interleukin-37 counteract this process, preserving mitochondrial homeostasis and limiting inflammation [232]. Glycerol-3-phosphate acyltransferase 4 (GPAT4), an endoplasmic reticulum–anchored enzyme, supports cardiac development by maintaining ER–mitochondrial communication and preventing mtDNA escape [233]. In human cardiac progenitor–like cells, argon preconditioning enhances cell membrane integrity and suppresses mtDNA release, protecting against oxygen–glucose deprivation–induced cell death [234].

Once in the cytosol or extracellular space, mtDNA can itself be targeted. Z-DNA binding protein 1 (ZBP1) binds released mtDNA to inhibit myocardial inflammation and adverse remodeling after MI, acting as an endogenous brake on DAMP signaling [160]. Intriguingly, in doxorubicin-treated cardiomyocytes, ZBP1 cooperates with cGAS to sustain type I interferon responses, illustrating its context-dependent dual roles in sensing mtDNA and modulating inflammation [235]. These findings underscore ZBP1’s multifunctionality, with its protective or pro-inflammatory effects determined by cell type and stimulus [160].

### 4.6. Regulation of Autophagy

When oxidative damage exceeds mitochondrial repair capacity, damaged mitochondria are selectively removed via autophagy, preventing accumulation of DAMPs such as mtDNA that would otherwise trigger sterile inflammation and release of pro-fibrotic cytokines [236,237]. In a heart-failure model induced by transverse aortic constriction, DNase II within autolysosomes degrades mtDNA fragments, protecting the myocardium from hemodynamic-stress-induced inflammation [141]. In ischemic myocardial injury, metformin suppresses inflammasome activation by enhancing macrophage autophagy, thereby degrading DAMPs and limiting myocardial injury [237]. Crizotinib—an ALK inhibitor used for metastatic non-small cell lung cancer—interrupts autophagosome-lysosome fusion, contributing to its cardiotoxicity; co-treatment with metformin restores autophagy and rescues crizotinib-induced cardiac injury [238]. Administration of β-hydroxybutyrate at the onset of reperfusion increases autophagic flux and maintains mitochondrial homeostasis by inhibiting the mTOR pathway [202]. Similarly, restoration of mitophagy with the inhibitor of MTORC1 can mitigate mitochondrial dysfunction and cardiomyopathy in Barth syndrome [239].

The PINK1/Parkin pathway is well-known as a mediator of mitophagy [240] and reasonably serves as a target for the treatment of cardiovascular diseases. PINK1 overexpression [144], omega-3 fatty acids [241], and vericiguat [145] protect the heart through promoting PINK1-mediated mitophagy and inhibiting the mtDNA release. On the other hand, astragaloside IV promotes the autophagy of angiotensin II-induced impaired mitochondria by facilitating the expression of Parkin and Drp1 [205].

Components of the cGAS-STING pathway, including cGAMP, cGAS, and TBK1, directly or indirectly induce autophagy and are essential for clearance of cytosolic DNA after genotoxic stress [157]. In Alzheimer’s disease–associated cardiac dysfunction, amyloid-β–induced reductions in melatonin impair mitochondrial aldehyde dehydrogenase activity, leading to mitochondrial damage and mtDNA release. Chronic cytosolic mtDNA accumulation then desensitizes cGAS-STING-TBK1 signaling, suppressing autophagy and further exacerbating amyloid-β and mitochondrial clearance deficits; melatonin supplementation reverses these effects by restoring aldehyde dehydrogenase activity, mitochondrial integrity, cGAS-STING-TBK1 signaling, and autophagic flux [242]. Paradoxically, in hypertension, mtDNA released from microglial mitochondria in the paraventricular nucleus activates cGAS, which in this context blocks autophagic flux and promotes neuroinflammation, sympathetic overdrive, and hypertensive cardiac injury [192]. We hypothesize that mtDNA-triggered, STING-dependent suppression of mitophagy forms a positive feedback loop in hypertensive heart disease, wherein STING activation impairs mitochondrial clearance, leading to further mtDNA release and sustained inflammation. Intracisternal infusion of the cGAS inhibitor RU.521 reverses these effects, underscoring the context-dependent interplay between mtDNA sensing and autophagy [192].

### 4.7. Ameliorating mtDNA-Triggered Inflammation

MtDNA released into the cytosol potently activates the cGAS–STING pathway, driving inflammatory cascades implicated in cardiovascular pathology. Consequently, inhibition of this axis has emerged as a promising therapeutic strategy. In murine models of high-fat diet–induced and aging-related atherosclerosis, the iridoid glycoside aucubin [243] and the phosphodiesterase-3 inhibitor cilostazol [244], respectively, attenuate lesion progression by suppressing STING signaling. STING deficiency likewise protects against diabetic cardiomyopathy and coxsackievirus B3–induced myocarditis [30,245]. In pulmonary arterial hypertension, cGAS–STING–NFκB activation by cytosolic mtDNA contributes to vascular remodeling; this is mitigated by calcitonin gene–related peptide administration [170]. In neuroinflammation-driven hypertensive cardiac injury, intracisternal infusion of the cGAS inhibitor RU.521 prevents mtDNA-mediated damage [192]. More recently, targeted disruption of cardiomyocyte–macrophage crosstalk has been achieved using a hydrogel patch delivering Rb1/PDA nanoparticles into the infarct border zone; these nanoparticles inhibit macrophage STING activation by cardiomyocyte-derived mtDNA, improving post-MI remodeling [20,246]. However, chronic blockade of cytosolic DNA sensing raises concerns about increased susceptibility to infection [247].

Released mtDNA also engages endosomal Toll-like receptors, particularly TLR9, initiating NFκB–dependent pro-inflammatory responses that promote NLRP3 inflammasome assembly and cytokine release [160,165]. In a SARS-CoV-2 myocarditis model, extracellular vesicles from human umbilical vein endothelial cells inhibit TLR-mediated NFκB activation, reducing myocardial inflammation and mitochondrial damage, and preserving cardiac function [134]. Similarly, TLR4 knockdown protects against circulating mtDNA–mediated inflammation in experimental autoimmune myocarditis [248].

Because the NLRP3 inflammasome mediates mtDNA-driven sterile inflammation, macrophage recruitment, and pyroptosis, its inhibition offers further therapeutic opportunities. The small-molecule inhibitor MCC950 [175], melatonin [249], and metformin [250] each suppress NLRP3 activation in macrophages or endothelial cells, attenuating atherosclerotic lesion development [162]. Electroacupuncture preconditioning decreases circulating mtDNA, prevents NLRP3 overactivation, and expedites the transition from pro-inflammatory to reparative phases after MI/RI [251]. In cardiac hypertrophy, the thrombin inhibitor hirudin reduces ROS, lowers cytosolic mtDNA, and inhibits NLRP3 assembly [193]. Upstream of inflammasome activation, enhancement of mitochondrial aldehyde dehydrogenase activity inhibits the caspase-1/GSDMD pyroptotic pathway, improving survival and preventing septic shock–associated cardiac dysfunction [252].

### 4.8. Others

Beyond the pathways detailed above, additional interventions targeting mtDNA-mediated mechanisms have demonstrated efficacy:•PCSK9 Inhibition. Proprotein convertase subtilisin/kexin type 9 (PCSK9) has been implicated in pyroptosis via mtDNA damage during chronic myocardial ischemia. Pharmacologic inhibition of PCSK9 reduces mtDNA lesions, attenuates NLRP3 activation, and confers cardioprotection in ischemic models [21,253]. Furthermore, PCSK9 inhibition exerts a lipid-lowering effect [254,255] and overcomes the limitations of statins in interfering with the attachment of mtDNA to the inner mitochondrial membrane [256], mtDNA depletion [257], suppressing coenzyme Q10 synthesis and inducing mtDNA oxidative damage [258,259], especially the lipophilic statins due to their non-selective diffusion to extrahepatic tissues [260].•Kearns–Sayre Syndrome Management. Kearns–Sayre syndrome, caused by large-scale mtDNA rearrangements, leads to progressive conduction system degeneration. Prophylactic implantation of a permanent pacemaker is often life-saving, preventing bradyarrhythmias and sudden cardiac death in affected patients [7,261].•Mitochondrial–Nuclear Exchange Models. In mitochondrial-nuclear exchange mice—where the mtDNA haplotype from C3H/HeN mice is placed on a C57BL/6J nuclear background and vice versa—mtDNA variations modulate nuclear gene expression, mitochondrial morphology, and function. Notably, the C3H mtDNA haplotype attenuates adverse remodeling in volume-overload heart failure, highlighting the protective role of specific mitochondrial genomes in cardiac stress responses [262].

**Table 2 antioxidants-14-01138-t002:** Summary of mtDNA-targeted therapeutic strategies and mechanisms.

Diseases	Treatments	Mechanisms
Hypertension	Calcitonin gene-related peptide [170]	Alleviating mitochondrial damage, mtDNA release, and cGAS-STING-NFκB activation
	RU.521 [192]	Inhibiting cGAS
	Ecklonia cava extract [182]	Decreasing mtDNA damage and mtROS generation
	Voluntary exercise [196]	Improving mtDNA integrity
	Oleuropein [207], Calcitriol [220]	Increasing mtDNA copy-number and regulating mitochondrial function
	MiRNA-21 [185]	Enhancing translation of mtDNA-encoded cytochrome b
Atherosclerosis	Chronic aerobic exercise [221]	Preserving mitochondrial function
	Ethidium bromide treatment [226]	MtDNA-depletion
	SRT1720 [210]	Improving mtDNA damage and mitochondrial function
	OGG1 [178]	DNA repair
	Aucubin [243], Cilostazol [132]	Suppressing STING signaling
	MCC950 [175], Melatonin [249], Metformin [250]	NLRP3 inhibition
Myocardial infarction	Astaxanthin [230], Octreotide [22]	Reducing oxidative damage, protecting mitochondria, and mtDNA
	Thymoquinone [231]	Inhibiting mtDNA loss, oxidative stress, inflammation, and apoptosis
	Dl-3-n-butylphthalide [217], SRT1720 [218], Twinkle overexpression [219]	Regulating mitochondrial function and biogenesis
	Metformin [237]	Alleviating autophagy-ROS-NLRP3 axis-mediated inflammatory response
	Rb1/PDA NPs-loaded hydrogel [246]	MtDNA-STING crosstalk modulation
	Deleting large tumor suppressor kinase 2 [18]	Preventing mtDNA release
Myocardial ischemia/reperfusion injury	Lycopene [194], Huoxue Huatan Decoction [195]	Restoring TFAM
	5-azacytidine [227]	Inhibiting DNMT1
	Tat-Beclin 1 [200], β-hydroxybutyrate [202]	Increasing autophagic flux
	Sappanone A [201]	Mitochondrial quality control
	Suberoylanilide hydroxamic acid [203]	Inducing autophagy and mitochondrial biogenesis
	Elevating OGG1 activity [26], Exscien1-III [189]	Base-excision repair
	Combination of Endo III with DNase I [26]	Repairing mtDNA and removing destroyed mtDNA fragments
	Epigallocatechin-3-gallate [228], Argon preconditioning [234]	Inhibition of mtDNA release
	Midkine, AS1411 [229]	Preventing mtDNA uptake
	Electroacupuncture preconditioning [251]	Reducing plasma mtDNA and modulating dynamic inflammatory response
	Suppressing proprotein convertase subtilisin/Kexin type 9 [21,253]	Partly through inhibition of pyroptosis via suppressing mtDNA damage
Diabetic cardiomyopathy	STING deficiency [30]	Inhibiting cardiomyocyte pyroptosis and inflammatory response
	Resveratrol [54]	Reducing TFAM acetylation
	IL-37 administration or IL-37 transgenosis [232]	Regulating mtDNA-enriched vesicle release
	Exercise training [180]	Reversing reduced mtDNA replication and transcription, and impaired mitochondrial ultrastructure
	A dominant negative *O*-GlcNAc transferase mutant F460A [186]	Restoring OGG1 enzymatic activity
Heart failure	TFAM overexpression [27,28,179]	Ameliorating decreased mtDNA copy-number and mitochondrial deficiencies [27,179] Inhibiting MMP9 protease expression and pathological cardiac remodeling [28]
	General control of amino acid synthesis 5-like 1 [197]	TFAM acetylation
	Twinkle overexpression [179,222]	Increasing mtDNA copy-number
	Mitochondrial topoisomerase 1 [223]	Maintaining mtDNA homeostasis
	Rectifying N^6^-methyladenine excess [40]	Inhibition of mtDNA methylation
	OGG1 overexpression [188], Exscien1-III [189]	Base-excision repair
	DNase II [141]	Digests mtDNA in the autophagy system
	Z-DNA binding protein 1 [160]	Suppressing mtDNA-induced inflammation
Myocarditis	Extracellular vesicles derived from human umbilical endothelial cells [134]	Inhibiting TLR-mediated NFκB activation
	STING deficiency [245]	Resist cardiac inflammation
	TLR4 knockdown [248]	Protecting against circulating mtDNA-mediated cardiac inflammation and injury
Cardiac hypertrophy	Recombinant human TFAM protein [199]	Increasing mtDNA and inhibiting nuclear factor of activated T cells
	Hirudin [193]	Inhibiting NLRP3 inflammasome formation
	PINK1 overexpression [144]	Ameliorating autophagy disturbance
	Radiotherapy [177]	Inducing oxidative stress, which causes mtDNA leakage and cGAS/STING/NLRP3-mediated pyroptosis
	Enalapril [72], Ablating lysocardiolipin acyltransferase 1 [204]	Attenuating mtDNA oxidative damage [72], and maintaining mitochondrial quality control [72,204]
Others	Recombinant human glucagon-like peptide-1 [224]	Preserving mtDNA content and mitochondrial biogenesis
	Prophylactic placement of a pacemaker [261]	Preserving heart conduction in Kearns-Sayre syndrome
	Twinkle [184], Astragaloside IV [205], omega-3 fatty acids [241]	Attenuating mtDNA oxidative damage [184,205], mediating autophagy [205,241]
	Oleoylethanolamide [209]	Attenuating mtDNA stress by activating PPARα
	5-aza-2′-deoxycytidine [122]	Inhibiting *COX2* gene methylation and downregulation
	C3Hmt haplotype [262]	Modulating genome expression and mitochondrial structure/function
	Metformin [238], Resveratrol [206], Vericiguat [145], Rapamycin [239]	Ameliorating mitophagy disturbance
	Melatonin [242]	Boosting ALDH2 activity
	Aldehyde dehydrogenase [252]	Inhibiting mitochondrion-NLRP3 pathway
	Glycerol-3-phosphate acyltransferase 4 [233]	Preventing mtDNA escape

## 5. Conclusions and Perspectives

In conclusion, mtDNA dysfunction constitutes a self-perpetuating vicious cycle at cardiovascular pathogenesis, mechanistically linking bioenergetic collapse to sterile inflammation. This cycle acts both as an instigator in susceptible individuals and an amplifier of injury in established disease. Disrupting this deleterious cycle requires a two-pronged therapeutic approach: alleviating mtDNA pathological characteristics and suppressing innate immune activation.

Despite these promising strategies, key questions remain unresolved. In some settings, mtDNA defects may be downstream consequences rather than primary causes. The heteroplasmy thresholds at which mutant mtDNA becomes pathogenic are poorly defined and most proposed therapies have not advanced beyond preclinical models. Rigorous mechanistic investigations and well-designed clinical trials will be essential to establish causality, identify at-risk patient populations, and translate this unified pathogenic framework into effective treatments.

Ultimately, future work must integrate mtDNA-focused diagnostics with targeted therapies to break this pathogenic cycle and improve cardiovascular outcomes.

## Figures and Tables

**Figure 1 antioxidants-14-01138-f001:**
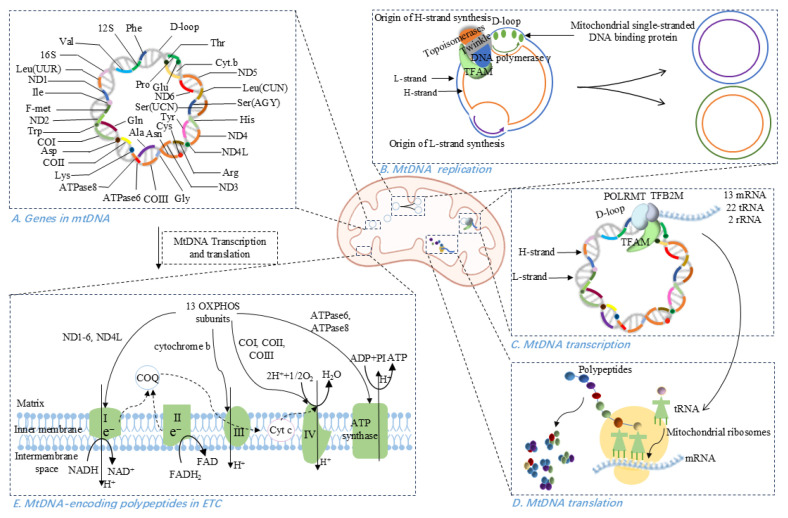
Physiological properties of mitochondrial DNA (mtDNA). (**A**) Gene distribution in mtDNA, with 28 genes on the heavy (H) strand and 9 on the light (L) strand. (**B**) MtDNA replication occurs in the ‘replisome’ comprising DNA polymerase γ, mitochondrial single-stranded DNA binding protein, Twinkle helicase, topoisomerases, and TFAM. (**C**) Transcription of mtDNA initiated at the D-loop region is regulated by POLRMT, TFAM, and TFB2M, producing 13 mRNAs, 22 tRNAs, and 2 rRNAs. (**D**) MtDNA-directed translation involves initiation complex assembly, mRNA binding to the ribosomal subunit, tRNA-mediated peptide elongation, translation termination, and ribosome recycling. (**E**) MtDNA-encoded polypeptides contribute to the function of ETC, specifically within complexes I, III, IV, and V.

**Figure 2 antioxidants-14-01138-f002:**
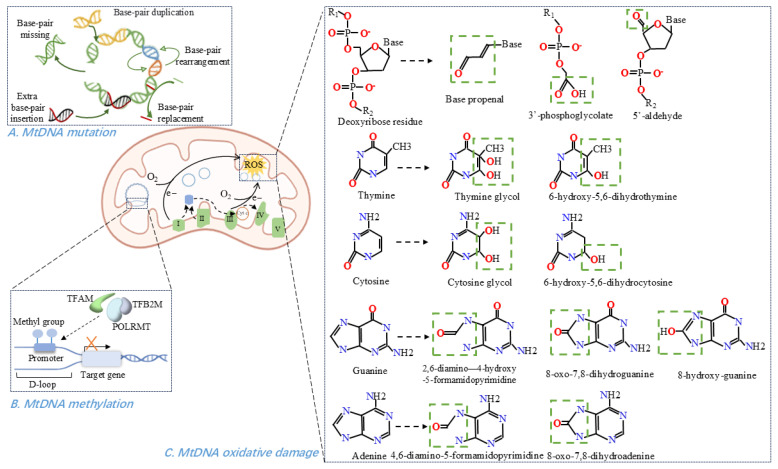
MtDNA-related pathological processes within mitochondria. (**A**) Forms of mtDNA mutations, include base-pair substitutions, rearrangements, duplications, deletions, and insertions. (**B**) Methylation of mtDNA in the D-loop region, involves methyl group addition and leads to transcriptional repression. (**C**) Representative oxidative modifications of mtDNA affect both the sugar-phosphate backbone and the four nucleotide bases.

**Figure 3 antioxidants-14-01138-f003:**
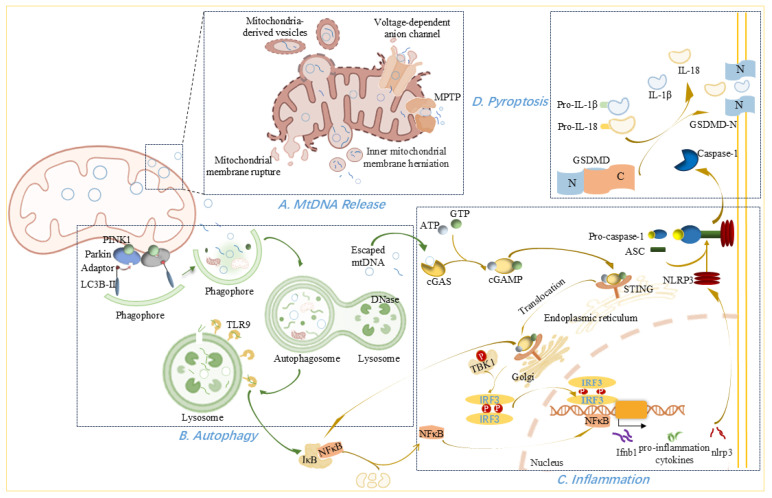
MtDNA release from mitochondria and its physiological and pathological effects outside the organelle. (**A**) Mechanisms of mtDNA release. MtDNA release occurs via multiple pathways: inner mitochondrial membrane herniation into the cytoplasm; oligomerization of VDACs forming outer membrane pores; leakage through the MPTP; mitochondria-derived vesicles; and excessive mitochondrial fission or rupture. (**B**) Autophagic Clearance of mtDNA. Autophagy begins with phagophore formation and expansion mediated by PINK1, Parkin, adaptor, and lipidated LC3B-II. The resulting autophagosomes subsequently fuse with lysosomes for mtDNA degradation. (**C**) mtDNA-Induced Inflammation. Cytosolic mtDNA activates innate immunity via TLR9–NFκB–NLRP3 signaling, leading to elevated inflammatory cytokines, and through the cGAS–cGAMP–STING axis, leading to TBK1/IKK-mediated IRF3 and NFκB activation. This promotes type I interferon and proinflammatory cytokine production (**D**) Pyroptosis pathway. NLRP3 inflammasome activation cleaves pro-caspase-1 into caspase-1, which processes pro-IL-1β and pro-IL-18 into active forms. Caspase-1 also cleaves GSDMD, generating the N-terminal fragment (GSDMD-N) that forms membrane pores, facilitating the secretion of IL-1β and IL-18 and leading to pyroptotic cell death.

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
