# Peer review of "Mitochondrial DNA Dysfunction in Cardiovascular Diseases: A Novel Therapeutic Target"

_antioxidants, 2025, doi:10.3390/antiox14091138_

Round 1

Reviewer 1 Report

Mitochondrial malfunction is a hallmark of many human diseases and the present review concentrates to cardiovascular diseases especially giving updated information on mtDNA dysfunction. The review also includes the current knowledge on  preclinical and clinical strategies that aim to improve the mtDNA function. As a whole the review is well written including figures and tables that further support readability of the paper. 

Mitochondria are an integral part of eukaryotic cells. They have their own genome, which in mammals encodes mitochondrial proteins of the electron transport chain (ETC) and harboring tRNAs and ribosomal subunit RNAs The rest of the mitochondrial-associated key proteins, including proteins of Krebs cycle and ETC complexes, are encoded by nuclear genes, and are imported into the mitochondria. Thus, signaling between the two organelles is necessary for mitochondrial integrity and cellular homeostasis. This signaling cascade is achieved through bidirectional communication between the nucleus and the mitochondria. The functional quality of mtDNA is of utmost importance. The present review well covers the major aspects of mtDNA function/malfunction clearly clarifying structural issues of mtDNA, replication, transcription and translation aspects, OXPHOS, mtDNA dysfunction associated with CVD, and at the end some therapeutic strategies targeted to mtDNA in the context of CVD. The cited literature is well updated including 235 references. There are some issues that should be considered in my following comments.

DETAILED COMMENTS

  1. Although the review majorily concentrates on mtDNA it is important to notify that mitochondrial quality control pathway also includes PINK1 - Parkin pathway whereby this pathway genes are also relevant to discuss with some details.
  2. In addition to maintain mitochondria beyond stress conditions. the BNIP3/Nix pathway is also important and the authors should add some basic points on its role here.
  3. The authors mention PCSK-9 as one factor related to initiation of mtDNA damage and also inducing pyroptosis. Its inhibition is currently considered a relevant tool to affect LDL levels. What is the effect of statin treatment in the context of mtDNA and since there are several statins some hydrophilic and some more hydriphobic, how do they affect mtDNA function?
  4. The authors should also comment whether high HDL level, high omega-3 fatty acids and high triglyceride levels affect mtDNA?  

Author Response

Reviewer 1

Major comments

Mitochondrial malfunction is a hallmark of many human diseases and the present review concentrates to cardiovascular diseases especially giving updated information on mtDNA dysfunction. The review also includes the current knowledge on preclinical and clinical strategies that aim to improve the mtDNA function. As a whole the review is well written including figures and tables that further support readability of the paper. 

Detailed comments

Mitochondria are an integral part of eukaryotic cells. They have their own genome, which in mammals encodes mitochondrial proteins of the electron transport chain (ETC) and harboring tRNAs and ribosomal subunit RNAs The rest of the mitochondrial-associated key proteins, including proteins of Krebs cycle and ETC complexes, are encoded by nuclear genes, and are imported into the mitochondria. Thus, signaling between the two organelles is necessary for mitochondrial integrity and cellular homeostasis. This signaling cascade is achieved through bidirectional communication between the nucleus and the mitochondria. The functional quality of mtDNA is of utmost importance. The present review well covers the major aspects of mtDNA function/malfunction clearly clarifying structural issues of mtDNA, replication, transcription and translation aspects, OXPHOS, mtDNA dysfunction associated with CVD, and at the end some therapeutic strategies targeted to mtDNA in the context of CVD. The cited literature is well updated including 235 references. There are some issues that should be considered in my following comments.

DETAILED COMMENTS

  1. Although the review majorily concentrates on mtDNA, it is important to notify that mitochondrial quality control pathway also includes PINK1 - Parkin pathway whereby this pathway genes are also relevant to discuss with some details.

Response1: We appreciate your suggestion. When discussing the mtDNA involved in mitochondrial quality control, we added information about the role of the PINK1 - Parkin pathway in mitophagy (In page 10, section 3.6, lines 405-415). And accordingly, we listed interventions against cardiovascular diseases targeting this signaling pathway (In page 20, section 4.6, lines 718-723).

  1. In addition to maintain mitochondria beyond stress conditions. the BNIP3/Nix pathway is also important and the authors should add some basic points on its role here.

Response 2: Thank you very much for your valuable suggestions. In our previous manuscript, we primarily focused on the role of autophagy in inhibiting mtDNA release or clearing free mtDNA after its release. However, based on your feedback, we have come to realize the reciprocal influence between mtDNA damage and impaired mitophagy. Therefore, we have supplemented the content related to BNIP3/Nix-mediated mitophagy to further elaborate the connection between mitophagy and mtDNA (In pages 10-11, section 3.6, lines 416-430).

  1. The authors mention PCSK-9 as one factor related to initiation of mtDNA damage and also inducing pyroptosis. Its inhibition is currently considered a relevant tool to affect LDL levels. What is the effect of statin treatment in the context of mtDNA and since there are several statins some hydrophilic and some more hydriphobic, how do they affect mtDNA function?

Response 3:Thank you for your valuable reminder. Through literature review, we have found that PCSK9 inhibitors indeed possess lipid-lowering effects and may serve as alternative treatments to statin-based lipid-lowering therapy to some extent. This is particularly relevant since statins, especially lipophilic statins, may affect the physiological properties of mtDNA, as detailed in the manuscript (In page 21, section 4.8, lines 782-787).

  1. The authors should also comment whether high HDL level, high omega-3 fatty acids and high triglyceride levels affect mtDNA? 

Response 4:Thank you for your comment. We found that HDL functions primarily as mediator of reversing cholesterol transport, followed by other effects including inhibition of mitochondrial permeability transition pore (mPTP) opening, preservation of mitochondrial function, suppression of inflammation, and protection against oxidative damage. Although direct evidence is still lacking, the properties of HDL may, to some extent, form a virtuous cycle with the normal functioning of mtDNA. (In page 18, section 4.2, lines 620-625)

Omega-3 fatty acids are reported as an enhancer of mitochondrial metabolism (In page 18, section 4.3, lines 644-647) and a promotor of PINK1-mediated mitophagy (In page 20, section 4.6, lines 719-721). However, the side effects caused by the overdosage of omega-3 fatty acids during pregnancy have also been reported [1].

To our knowledge, elevated triglyceride levels predominantly exert negative effects on mtDNA integrity and function, and these effects are not exerted in isolation but rather through dynamic interconversion and regulatory interactions with various metabolic intermediates. As highlighted by the reviewer, the effects of fatty acids, triglycerides, and other related metabolites—as well as their dysregulation on mtDNA and heart—have been discussed in detail in the manuscript. (In page 10, section 3.5, lines 378-388)

References

  1. Hsueh, T.Y., J.I. Baum, and Y. Huang, Effect of Eicosapentaenoic Acid and Docosahexaenoic Acid on Myogenesis and Mitochondrial Biosynthesis during Murine Skeletal Muscle Cell Differentiation. Front Nutr, 2018. 5: p. 15.

Reviewer 2 Report

No novelty of this review. Please think and gather important knowledge. 

This review article investigates preclinical and clinical strategies aimed at interrupting mtDNA cycle—bolstering mtDNA repair and copy-number maintenance, reversing pathogenic methylation, and blocking mtDNA-triggered innate immune activation. This is a valuable topic; however, several concerns need to be addressed:

  1. Title is not clear please modify it.
  2. Why is there no aim in the abstract? Or why is there no stated purpose?
  3. The key conclusion is not clearly defined, either in the abstract or within the main manuscript. Please revise it to ensure that the central takeaway of the study is explicitly stated and easily understood by readers.
  4. The introduction would benefit from a more comprehensive review of previous studies investigating the relationship between Mitochondrial DNA and Cardiovascular Diseases.
  5. Please have the manuscript reviewed for grammatical accuracy by a native speaker.
  6. How do you draw the figure 1 and others ?
  7. line 216 why big gap ?
  8. Line 377 need ref.
  9. For the Autophagy, inflammation and pyroptosis, you need to draw a figure and mentioned the possible mechanism, how it involve ?
  10. Repairs of mtDNA Oxidative Damage ? how can it involve in cardiovascular diseases ?
  11. Autophagy and Regulating Autophagy ? what is the difference ?
  12. I did not find any Targeted Therapies ? or any clinical study ? there is no co-relationship with your title.
  13. I feel sometimes you just increase your writing word number without thinking or without any novel information. For example, Ameliorating mtDNA Triggered Inflammation and another paragraph just inflammation. Please concise it.
  14. to be honest there is no novelty of this manuscript. Just add the all the published paper data. Actually, I am not sure what is the novelty of this review. Could you please explain ? I found everything already online.
  15. Make some new hypothesis.
  16. With hypothesis please make more evidence to support your idea. Generate some new knowledge.
  17. There are inappropriate uses of periods throughout the manuscript. Please review and revise punctuation.
  18. Verb tenses should be consistent—ideally, use the past tense throughout.

Author Response

Major comments

No novelty of this review. Please think and gather important knowledge. 

Detailed comments

This review article investigates preclinical and clinical strategies aimed at interrupting mtDNA cycle—bolstering mtDNA repair and copy-number maintenance, reversing pathogenic methylation, and blocking mtDNA-triggered innate immune activation. This is a valuable topic; however, several concerns need to be addressed:

  1. Title is not clear please modify it.

Response 1: We thank the reviewer for valuable feedback. In response to their suggestions, the title of the manuscript has been revised from Mitochondrial DNA in Cardiovascular Diseasesfrom Cellular Biology to Targeted Therapies to Mitochondrial DNA Dysfunction in Cardiovascular Diseases: A Novel Therapeutic Target. While the original version provided a concise overview of the physiological characteristics of mtDNA, the new title more directly highlights “dysfunction” as the central theme and emphasizes its role as a therapeutic target in cardiovascular diseases.

  1. Why is there no aim in the abstract? Or why is there no stated purpose?

Response 2: Thank you for your keen and professional suggestion. In the abstract of this review, we opted for a distinct structural approach. To present the purpose more explicitly and clearly, we have made brief revisions as highlighted in red. The phrase " In this review, we aim to synthesize " (In page 1, lines 28-29) and the concluding sentence " Furthermore, we seek to evaluate… and discuss…"(In page 1, lines 29-34) were intended to convey our aims indicating that the article would overview mechanisms, review certain strategies and discuss specific challenges.

  1. The key conclusion is not clearly defined, either in the abstract or within the main manuscript. Please revise it to ensure that the central takeaway of the study is explicitly stated and easily understood by readers.

Response 3: Thank you for the reviewer's valuable suggestion. We agree that the previous conclusion is more like a general summary of the manuscript rather than an unequivocal final statement. In response to the reviewer's comment, we have revised both the Abstract and the Conclusion sections.

In the Abstract, we have added the conclusion that the mtDNA-triggered pathological cascade in the manuscript is representative in cardiovascular diseases, i.e. We propose that this cycle represents an almost unifying pathogenic mechanism in a spectrum of mtDNA-driven cardiovascular disorders (In page 1, lines 26-27).

In the revised Conclusion section, we first present a synthesized paragraph summarizing the role of mtDNA dysfunction in the pathogenesis and treatments of cardiovascular diseases (In page 24, section 5, lines 800-805). This is followed by a discussion of limitations and challenges in current researches (In page 24, section 5, lines 806-812). Finally, we outline future directions concerning mtDNA-focused mechanisms and their therapeutic implications for cardiovascular diseases (In page 24, section 5, lines 813-814).

We hope that these revisions to the Abstract and Conclusion now successfully highlight the central takeaway of our manuscript.

  1. The introduction would benefit from a more comprehensive review of previous studies investigating the relationship between Mitochondrial DNA and Cardiovascular Diseases.

Response 4: Thank you for the reviewer's reminder. The studies on mtDNA and cardiovascular diseases have been summarized in the introduction. Please refer to the highlight in red. (In page 2, section 1, lines 55-67)

  1. Please have the manuscript reviewed for grammatical accuracy by a native speaker.

Response 5: Thank you for your kind reminder. Among the co-authors of this study, Professor Alexey Sarapultsev is a native English speaker. He has reviewed and polished the manuscript prior to its initial submission.

  1. How do you draw the figure 1 and others?

Response 6: Thank you for your inquiry. The figures in this manuscript were primarily created using PowerPoint. For certain complex elements, we sourced materials from the official BioRender website (https://www.biorender.com).

  1. line 216 why big gap?

Response 7: Thank you for kind reminder. In Microsoft Word, when editing a long word or terms connected by "/" (such as "2,6-diamino-4-hydroxy-5-formamidopyrimidine/4,6-diamino-5-formamidopyrimidine" now in page 6, section 3.1, lines 209-210), the program treats them as a single word or phrase by default. When full justification is applied, Microsoft Word prioritizes adjusting the width of spaces between words. If a line contains particularly long terms, the spacing between words may be excessively stretched, resulting in visibly irregular gaps. Therefore, this is normal, stemming from Microsoft Word 's justification algorithm and hyphenation rules, and can be resolved during later typesetting stages.

  1. Line 377 need ref.

Response 8: Thank you for your careful reminder. In line 377 and the two surrounding lines of the original manuscript the VDAC1 was described: “Under oxidative stress, voltage-dependent anion channel 1 (VDAC1)—the most abundant outer-membrane protein, which regulates Ca²⁺ influx, metabolism, inflammasome activation, and cell death—oligomerizes to create large pores that facilitate mtDNA release [130].” (In page 9, section 3.5, lines 352-355) The corresponding reference is provided at the end of this sentence: [130] Kim, J.; Gupta, R.; Blanco, L.P.; Yang, S.; Shteinfer-Kuzmine, A.; Wang, K.; Zhu, J.; Yoon, H.E.; Wang, X.; Kerkhofs, M., et al. VDAC oligomers form mitochondrial pores to release mtDNA fragments and promote lupus-like disease. Science. 2019, *366*, 1531–1536. https://doi.org/10.1126/science.aav4011

  1. For the Autophagy, inflammation and pyroptosis, you need to draw a figure and mentioned the possible mechanism, how it involve?

Response 9: We appreciate your suggestion. In the original manuscript, Figure 3 illustrates that released mtDNA is initially controlled by autophagy. Some mtDNA that escapes autophagic clearance subsequently triggers inflammatory pathways and ultimately leads to inflammatory cell death, specifically pyroptosis.

  1. Repairs of mtDNA Oxidative Damage? how can it involve in cardiovascular diseases?

Response 10: Thank you for your question. In Section 4.1. “Repair of mtDNA Oxidative Damage” the repair of mtDNA oxidative damage and its involvement in the treatment of cardiovascular diseases have been systematically elaborates.

It begins by reaffirming that the unique location and structure of mtDNA make it highly vulnerable to oxidative damage, emphasizing that such damage can lead to cardiovascular pathologies (the relevant mechanisms have been detailed in Section 3.1. “mtDNA Oxidative Damage”). Subsequently, this section focuses on providing therapeutic evidence for repair strategies at multiple levels: At the genetic level, overexpression of Twinkle enhances the capacity for oxidative damage repair, thereby protecting against cardiomyopathy. Meanwhile, miRNA-21 upregulates the translation of cytochrome b, reduces ROS production, and ameliorates hypertension-related cardiac pathology. In terms of pharmacological interventions, Ecklonia cava extract and oleoylethanolamide mitigate mtDNA oxidative damage, alleviating vascular calcification and cellular senescence. The most compelling evidence comes from targeted interventions involving repair enzymes—enhancing OGG1 activity and administering Exscien1-III directly demonstrate that delivering repair enzymes themselves constitutes an effective therapeutic strategy. Finally, this section also discusses the current limitations of repair strategies (e.g., the potential inadequacy of the base excision repair pathway alone in coping with severe ROS bursts) and proposes combination therapies (such as Endo III combined with DNase I). Through the synergistic effects of repairing damage and clearing irreversibly damaged mtDNA fragments, these strategies collectively enhance cardioprotection.

  1. Autophagy and Regulating Autophagy? what is the difference?

Response 11: Thank you for the reviewer's thoughtful question. Although both the "Autophagy" and "Regulation of Autophagy" sections focus on autophagy and mitochondrial DNA (mtDNA), they differ significantly in their core emphasis, perspective, and function. To put it simply, section "3.6 Autophagy" serves as an "introduction to fundamental knowledge". It primarily explains what autophagy (especially mitophagy) is, how it works, and its basic relationship with mtDNA release and inflammation. Section "4.6 Regulation of Autophagy" serves as a "discussion on application and regulation". It explores how interventions can be used to modulate the autophagy process to enhance mitochondrial renewal function, clear released mtDNA, or ameliorate cardiovascular diseases.

In summary, Section 3.6 "Autophagy" establishes the mechanistic foundation, providing the necessary theoretical support for the "therapeutic applications" discussed in Section 4.6 "Regulation of Autophagy".

  1. I did not find any Targeted Therapies? or any clinical study? there is no co-relationship with your title.

Response 12: Thank you for raising the question.

In Section "4. Therapeutic Strategies Targeting mtDNA and Its Related Pathways in Cardiovascular Diseases", various preclinical interventional strategies targeting mtDNA for the treatment of cardiovascular diseases are elaborated according mtDNA lesions itself, mtDNA release, and the inflammation and cell death triggered by free mtDNA. Most of these strategies are currently at preclinical stage, while direct evidence from human clinical studies remains relatively limited, as discussed in the Conclusion.

The following is a summary of how these interventions relate to the title "Therapeutic Strategies Targeting mtDNA and Its Related Pathways in Cardiovascular Diseases": (1) Strategies directly targeting mtDNA itself, such as enhancing OGG1 activity and using the fusion peptide Exscien1-III, aim to protect, repair, or maintain mtDNA integrity, thereby preventing damage at its source. (2) Maintaining mtDNA quality and copy number, for example, by protecting mtDNA integrity through TFAM and promoting mtDNA biogenesis. (3) Regulating mtDNA epigenetics to alleviate abnormal post-transcriptional modifications of mtDNA, thereby preventing its detrimental effects on the cardiovascular system. (4) Strategies targeting mtDNA release, which seek to "lock" mtDNA within mitochondria, preventing it from acting as a DAMP to initiate inflammation. (5) Clearing mtDNA via autophagy/mitophagy, a strategy that primarily enhances the cell's own "waste disposal system" to degrade released mtDNA. (6) Strategies targeting inflammation pathways triggered by mtDNA, which come into play after mtDNA has been released, mitigating inflammatory damage by inhibiting immune signaling pathways activated by mtDNA.

Together, we aimed to provide a systematic elaboration about interventions as a comprehensive system for "Targeting mtDNA and Its Related Pathways in Cardiovascular Diseases": from preventing damage, to controlling release, to neutralizing inflammation, and finally to clearing the aftermath.

  1. I feel sometimes you just increase your writing word number without thinking or without any novel information. For example, Ameliorating mtDNA Triggered Inflammation and another paragraph just inflammation. Please concise it.

Response 13: Thank you very much for pointing this out. Our review aims to summarize the fundamental biology of mtDNA, with a particular focus on that mtDNA dysfunction—primarily by undermining cardiac bioenergetics and unleashing sterile inflammation—initiates or amplifies cardiovascular diseases. We have sought to provide an integrative framework that categorizes therapeutic strategies into distinct levels of intervention: inhibiting mtDNA damage, preventing mtDNA release, counteracting inflammation, and promoting mtDNA clearance. It is our hope that this framework offers a novel perspective on the treatment of cardiovascular diseases.

We acknowledge that, in our initial draft, the effort to be comprehensive may have resulted in certain sections being overly descriptive at the expense of critical analysis. In response to your feedback, we will refine the manuscript by removing redundant statements and ensuring more concise and focused writing.

Regarding the sections "Inflammation" and "Ameliorating mtDNA-Triggered Inflammation" cited by the reviewer, similar to the previously described relationship between "Autophagy" and "Regulation of Autophagy", the fundamental mechanisms and interventional strategies were described sequentially in these two sections.

  1. to be honest there is no novelty of this manuscript. Just add the all the published paper data. Actually, I am not sure what is the novelty of this review. Could you please explain? I found everything already online.

Response 14: We thank the reviewer for this critical comment。We believe that the "novelty" of a review article generally does not lie in reporting unpublished data, but rather in integrating fragmented knowledge into a coherent conceptual framework, re-examining established questions from an underexplored perspective, and critically analyzing "what remains unknown," "contradictions and limitations in current research," while also providing guidance for future studies.

While individual findings shown in this manuscript are available online, a key novelty of our review is that we propose a new ‘Damage-Control’ framework for understanding mtDNA-driven cardiovascular pathologies. We categorize existing interventions not by disease, but by their point of action in the pathological cascade: (1) preventing mtDNA damage, (2) blocking its release, (3) neutralizing its inflammatory signaling, and (4) promoting its clearance. This integrative perspective provides a more actionable and mechanistic understanding of the field, which to our knowledge, has not been previously presented in this manner.”

  1. Make some new hypothesis.

Response 15: We thank the reviewer for this exceptional suggestion. While traditional review articles typically synthesize existing knowledge, we agree that proposing novel, testable hypotheses can elevate a review’s impact by providing a forward-looking perspective for the field. Therefore, we formulated a set of mechanism-driven, speculative hypotheses based on the current literature, intending to bridge known facts with unknown mechanisms and are presented as a springboard for future experimental validation.
(1) We hypothesize that the combined treatment, such as DNase I +Endo III, targeting mtDNA damage and the pathological cascade it triggers is universal: Therefore, such combination therapy targeting both mtDNA damage and its immunostimulatory properties, alongside other therapeutic strategies, is expected to exhibit broad applicability in cardiovascular protection. (In page17, paragraph 4.1, lines 587-590)

(2) Regarding the issue of STING inducing autophagy in the ER-Golgi intermediate compartment, we propose the hypothesis: We hypothesize that STING activation at the ER-Golgi intermediate compartment stimulates the local production of autophagy-specific phosphoinositides or facilitates the transfer of lipids to create a lipid microenvironment conducive to the elongation and curvature of phagophores. (In page11, paragraph 3.6, lines 450-453)

(3) In the context that activated cGAS blocks autophagic flux and promotes neuroinflammation, we hypothesize: We hypothesize that mtDNA-triggered, STING-dependent suppression of mitophagy forms a positive feedback loop in hypertensive heart disease, wherein STING activation impairs mitochondrial clearance, leading to further mtDNA release and sustained inflammation. (In page 20, paragraph 4.6, lines 735-738)

  1. With hypothesis please make more evidence to support your idea. Generate some new knowledge.

Response 16: Thank for the reviewer’s kind suggestion.

(1) Regarding the hypothesis of combination therapies targeting different stages of mtDNA pathogenesis, interventions in the context of hypertension have been cited as examples. Please refer to the revised manuscript for details. (In page 17, paragraph 4.1, lines 607-611)

(2) The hypothesis regarding STING activation at the ER–Golgi intermediate compartment stimulating autophagy is supported by evidence presented in the manuscript. (In page11, paragraph 3.6, lines 470-473)

(3) For the hypothesis that mtDNA-triggered, STING-dependent suppression of mitophagy forms a positive feedback loop in hypertensive heart disease, there is already supporting evidences from existing literature in the manuscript:
① Damaged mitochondria can be cleared through autophagy: When oxidative damage exceeds mitochondrial repair capacity, damaged mitochondria are selectively removed via autophagy…. (In page19, paragraph 4.6, lines 703-706)

②Damage mitochondrial releases mtDNA, activating cGAS-STING: Myocardial mitochondrial impairment, mtDNA release, and subsequently STING/p65 activation. (In page19, paragraph 4.5, lines 677-678)

③cGAS inhibits mitophagy: Paradoxically, in hypertension, mtDNA released from microglial mitochondria in the paraventricular nucleus activates cGAS, which in this context blocks autophagic flux and promotes neuroinflammation. (In page20, paragraph 4.6, lines 732-735)

  1. There are inappropriate uses of periods throughout the manuscript. Please review and revise punctuation.

Response 17: Thank you for your kind reminder. We have checked and revised the inappropriate punctuation in the manuscript.

  1. Verb tenses should be consistent—ideally, use the past tense throughout.

Response 18: Thank you for this meticulous comment. We have now carefully revised the entire manuscript to ensure that the past tense is used consistently when describing the findings from previously published studies.

Reviewer 3 Report

This review provides a compelling and comprehensive overview of the role of mitochondrial DNA (mtDNA) dysfunction in the pathogenesis of cardiovascular diseases. It successfully integrates molecular biology, immunology, and clinical relevance into a unified framework that highlights the dual role of mtDNA as both a trigger and amplifier of disease.

This review offers substantial mechanistic insight, particularly in its detailed analysis of mtDNA release mechanisms and the downstream activation of inflammatory pathways, including the cGAS–STING axis and the NLRP3 inflammasome.

This review provides a compelling and comprehensive overview of the role of mitochondrial DNA (mtDNA) dysfunction in the pathogenesis of cardiovascular diseases. It successfully integrates molecular biology, immunology, and clinical relevance into a unified framework that highlights the dual role of mtDNA as both a trigger and amplifier of disease.

This review offers substantial mechanistic insight, particularly in its detailed analysis of mtDNA release mechanisms and the downstream activation of inflammatory pathways, including the cGAS–STING axis and the NLRP3 inflammasome.

Author Response

Major comments

This review provides a compelling and comprehensive overview of the role of mitochondrial DNA (mtDNA) dysfunction in the pathogenesis of cardiovascular diseases. It successfully integrates molecular biology, immunology, and clinical relevance into a unified framework that highlights the dual role of mtDNA as both a trigger and amplifier of disease.

This review offers substantial mechanistic insight, particularly in its detailed analysis of mtDNA release mechanisms and the downstream activation of inflammatory pathways, including the cGAS–STING axis and the NLRP3 inflammasome.

Detailed comments

This review provides a compelling and comprehensive overview of the role of mitochondrial DNA (mtDNA) dysfunction in the pathogenesis of cardiovascular diseases. It successfully integrates molecular biology, immunology, and clinical relevance into a unified framework that highlights the dual role of mtDNA as both a trigger and amplifier of disease.

This review offers substantial mechanistic insight, particularly in its detailed analysis of mtDNA release mechanisms and the downstream activation of inflammatory pathways, including the cGAS–STING axis and the NLRP3 inflammasome.

Response: Thank you very much for your recognition of our work and for such positive feedback. You have accurately summarized the core objective of our review—to establish a unified framework elucidating how mtDNA dysfunction drives the clinical pathology of cardiovascular diseases from a molecular perspective. We are very pleased that you found the review successful in connecting molecular mechanisms with clinical relevance and in highlighting the central framework of mtDNA as both a "trigger and amplifier." This is precisely the key message we aimed to convey to our readers.

Round 2

Reviewer 1 Report

The authors have responded to all of my queries in a very good way and I do not have any further comments.

No further detailed comments.

Author Response

Major comments

The authors have responded to all of my queries in a very good way and I do not have any further comments.

Detailed comments

No further detailed comments.

Response: We are very pleased to hear that the reviewer is satisfied with our revisions and have no further comments. We thank you for your time and constructive feedback, which have significantly improved our manuscript.

Reviewer 2 Report

minor

It is now better than previous writing but still missing some information. Such as the mechanism of mtDNA or copy of mtDNA. How it involve in cardiovascular ? please explain. 

Author Response

Major comments

minor

Detailed comments

It is now better than previous writing but still missing some information. Such as the mechanism of mtDNA or copy of mtDNA. How it involve in cardiovascular ? please explain.

Response: We sincerely thank the reviewer for this insightful comment. In Section 3.3, we briefly introduce the general pathological characteristics of mtDNA copy reduction and compensatory increase (respectively in paragraphs 1 and 2 of this section), followed by contextualizing its relevance to cardiac mitochondrial function and cardiovascular diseases (in paragraph 3 of this section). Accordingly, in Section 4.3, we summarize corresponding interventions targeting mtDNA copy number variation, with the aim of treating cardiovascular diseases by maintaining mtDNA content.